



# Novel clustering framework using *k*-means (S *k*-means) for mining spatiotemporal structured climate data

Quang-Van Doan[1], Toshiyuki Amagasa[1], Thanh-Ha Pham[2], Takuto Sato[1], Fei Chen[3], Hiroyuki Kusaka[1]

[1]Center for Computational Sciences, University of Tsukuba, Japan
[2]Hanoi University of Sciences, National University Hanoi, Vietnam
[3]Research Applications Laboratory, National Center for Atmospheric Research, USA

*Correspondence to*: Quang-Van Doan (doan.van.gb@u.tsukuba.ac.jp)

**Abstract.** Dramatic increases in climate data underlie a gradual paradigm shift in knowledge-acquisition methods from physical-based models to data-based mining techniques. *k*-Means is one of the most popular data clustering/mining techniques,
and it has been used to detect hidden patterns in climate systems. *k*-Means is established based on distance metrics for pattern recognition, which is relatively ineffective when dealing with "structured" data that are dominant in climate science, that is, data in time and space domains. Here, we propose (i) a novel structural similarity recognition-based *k*-means algorithm called structural *k*-means or S *k*-means for climate data mining and (ii) a new clustering uncertainty representation / evaluation framework based on the information entropy concept. We demonstrated that the novel S *k*-means could provide higher-quality
clustering outcomes in terms of general silhouette analysis, although it requires higher computational resources compared with conventional algorithms. The results are consistent with different demonstration problem settings using different types of input data, including two-dimensional weather patterns, historical climate change in terms of time series, and tropical cyclone paths. Additionally, by quantifying the uncertainty underlying the clustering outcomes we for the first time evaluated the "meaningfulness" of applying a given clustering algorithm for a given dataset. We expect that this study will constitute a new
standard of *k*-means clustering with "structural" input data, as well as a new framework for uncertainty representation/evaluation of clustering algorithms for (but not limited to) climate science.

## 1. Introduction

In recent decades, the volume and complexity of climate data have increased dramatically owing to advancements in data acquisition methods (Overpeck et al., 2011). This increase underlies a gradual shift of climate-knowledge acquisition paradigm





from using classical "first-principle" models (i.e., based on physical laws) to models and analyses directly based on data (i.e., based on data mining) (Kantardzic, 2011). Hence, numerous data mining methods have been developed to investigate the underlying nature and structure of data. Clustering is one of the principal unsupervised data mining methods. It is used to organize a set of data into clusters that maximize the homogeneity of the elements in a cluster and the heterogeneity among different clusters (Pérez-Ortega et al., 2019). Clustering algorithms are useful to handle large, multivariate, and highly

dimensional data, which are difficult for human perception. Although numerous clustering algorithms exist, $k$-means is one of the most well-known and widely used in most research domains (Wu et al., 2008).

The history of $k$-means can be traced back to the 1950s – 1960s, when it was developed through independent efforts (e.g., Lloyd, 1957; Forgy, 1965; Jancey, 1966; MacQueen, 1967). The name $k$-means was coined in a paper by MacQueen (MacQueen, 1967). $k$-Means has been extensively used in climate science, thanks to its ease of implementation and

interpretation. It is used to explore unknown atmospheric mechanisms and/or improve predictions. The most common application is the use of $k$-means within "detection-and-attribution" framework. In this framework, first, specific atmospheric conditions, or events, e.g., abnormally hot weather or heavy precipitation, were detected. Then, the causes of these atmospheric conditions are attributed to atmospheric regimes/patterns, determined by using $k$-means (Esteban et al., 2005; Houssos et al., 2008; Spekat et al., 2010; Zeng et al., 2019; Smith et al., 2020). A different application is the use of $k$-means for weather or

climate predictions. In such a case, rather than being used as an independent prediction method, it is considered as complementary to existing numerical predictions. The combined $k$-means and numerical forecast system suggests the probability of occurrence of a certain weather condition by searching $k$-means derived analog patterns from historical data (Kannan and Ghosh, 2011; Gutiérrez *et al.*, 2013; Le Roux *et al.*, 2018; Pomee and Hertig, 2022). On top of this, under the analog approach, the algorithm can be used for future climate prediction (also known as a statistical downscaling) or for

reconstructing historical data (Camus et al., 2014).

The $k$-means algorithm is an interactive clustering method. To briefly describe, it involves four processing steps: $i$) initiation: predefines $k$ cluster centers (or centroids); $ii$) classification: clustering of an object with similar objects; $iii$) centroid update: recalculates centroids based on the updated classification; $iv$) convergence (equilibrium) judgement: halts the algorithm if object migrations are not observed from one cluster to another; otherwise, returns to step ii) if such migrations are observed

(Pérez-Ortega et al., 2019). The dominance of $k$-means over most research fields is partly due to its simplicity and ease of use. However, simplicity inherits the drawbacks of the algorithm. Such drawbacks (which will be explained later) have inspired



researchers for decades to identify improvements. Consequently, these efforts have delivered a great number of *k*-means variants alongside those from the earliest time.

Improving centroid initialization represents an important issue to be resolved. *k*-Means clustering outcomes are known to be sensitive to the initialization of centroids (Sydow, 1977; Katsavounidis et al., 1994; Bradley and Fayyad, 1998; Pelleg, 2000; Khan and Ahmad, 2004; Arthur and Vassilvitskii, 2006; Su and Dy, 2007; Eltibi and Ashour, 2011). Subsequent efforts have been made to improve the calculation procedure in the classification scheme primarily because it is the most computationally time-consuming. These efforts resulted in numerous *k*-means variants (Fahim et al., 2006; Lai and Huang, 2010; Perez et al., 2012). More recent studies have focused on the underlying basis of the classification, that is, how to define the similarity for which an object should be classified to one cluster but not another.

The conventional *k*-means classification scheme is established based on the distance paradigm, in which the similarity is determined by distance metrics. Such metrics include the Euclidean distance, Manhattan distance, or their general form, the Minkoski distance (Cordeiro de Amorim and Mirkin, 2012). The advantage of distance metrices lies in their ease of implementation and popularity, thus making the judgement for using them less controversial. Nonetheless, recent studies have pointed out that distance metrics defend less against noisy and irrelevant features (or dimensions, in other words) of input objects (vectors) (de Amorim, 2016). Few studies have proposed the use of feature weights to overcome this weakness (Chan et al., 2004; Huang et al., 2005; Cordeiro de Amorim and Mirkin, 2012). However, such improvements do not intentionally consider the structural relationship between vector dimensions, especially when data are time series or spatially distributed.

Atmospheric data are characterized by their temporal and spatial "structuredness". In other words, the information value of data lies in their relationship or trends in time and space. For example, when looking at weather maps, one might realize that locations of high or low pressures would be the first concern. The similarity, trend, or phase correlation of a time series might be more important than their absolute values. For these reasons, *k*-means under the distance paradigm treats the features of the input data equally, thus mask the similarity recognition between data, consequently deteriorating the clustering outcomes. This is true at least for a specific case of atmospheric data. However, to replace distance metrics by something different remains big challenging. It is because distance metrics have deep historical roots, and they undoubtedly laid the foundation for modern data mining, including clustering algorithms. As mentioned by Wang et al., "it (distance metric) is not bad, and easy to use" and "everyone else use it" (Wang et al., 2004).





Based on the nature of atmospheric data, a specific question raised here is whether another *k*-means approach is available that can consider the "structural" similarity in time and space between input objects. Answering this question has great practical value, particularly, for the climate informatics field owing to the unprecedented recent increase in archived data. The demand is growing for innovative and effective tools of data mining that can handle the inherent nature of climate data.

Here we propose a novel *k*-means algorithm based on the "structural" similarity recognition, called *Structural k-means* or *S k-means*. S *k*-means follows the same procedure as the generic *k*-means algorithm. It differs from the generic algorithm in incorporating a recent innovation in signal processing science, namely, the structural similarity (S-SIM) recognition concept (Wang et al, 2004) into the classification scheme. The novel S *k*-means inherits the simplicity of the generic algorithm and meanwhile can handle temporally and spatially ordered data.

We evaluate the performance of S *k*-means clustering across three representative demonstration tests. The tests cover multiple types of input data, that is, spatial distributions (weather patterns), time series (historical change in temperature), and hybrid types (tropical cyclone tracking). Using multiple data types is a unique point of this study that can robustify the conclusions through cross comparisons. The performance of S *k*-means is evaluated against three other *k*-means algorithms using different similarity/distance metrics for the classification scheme, that is, the Pearson correlation coefficient, Euclidean, and Manhattan distances, hereafter called C, E, and M *k*-means, respectively. We implement various *k* (number of centroids) configurations and multiple initializations (randomized). Eventually, 1320 model runs are conducted. Such settings ensure the robustness of the results and conclusions. The "general" silhouette analysis/score, which is a scoring method based on general similarity/distance metrics, is used to quantify the algorithm performance.

We propose a novel framework for *clustering-uncertainty evaluation/representation* based on the information entropy concept. This framework is primarily used to quantify the variability/consensus among the clustering outcomes across the different *k*-means algorithms. At the core of the framework is the newly proposed concept "*clustering uncertainty degree*," which builds on mutual information theory. Also, relevant visualization tools including connectivity matrix, heatmap, and chord diagram are proposed to represent the clustering uncertainty.

To the best of our knowledge, this study is the first to address the uncertainty issue in the climate science. Our study is the first to propose a clustering-uncertainty evaluation framework, borrowing recent-most techniques and concepts in information theory. This framework is not only used to quantify the clustering uncertainty but also to serve a more fundamental purpose, i.e., to measure the "meaningfulness" of the application of clustering for a given problem dataset. We expect that this

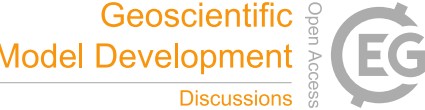

framework together with the S $k$-means algorithm will establish a new standard in data mining and clustering studies, primarily for (but not limited to) climate science.

The remainder of this paper is organized as follows. Section 2 describes the S $k$-means algorithm. Section 3 presents the test simulation configurations. Section 4 describes the evaluation metrics and a novel framework for clustering uncertainty. Section 5 presents and discusses the results. Section 6 provides the concluding remarks.

**2. Description of the algorithms**

**2.1 S $k$-means algorithm**

S $k$-means follows the conventional procedure of generic $k$-means clustering. To express this mathematically, let define $X = \{x_1, \ldots x_i, \ldots, x_n\}$ be a set of $n$ objects (input vectors), where $x_i \in R^d$ ($i = 1, \ldots, n$) and $d \geq 1$ is the number of dimensions. Let $K = \{1, \ldots, k\}$ with $k \geq 2$ denote the number of groups.

For a $k$-partition, $P = \{G(1), \ldots, G(k)\}$ of $X$, let $c_j$ denote the centroid of cluster $G(j)$, for $j \in K$, with $C = \{c_1, \ldots, c_k\}$ and a set of weight vectors $W = \{w_{11}, \ldots, w_{ij}\}$. Hence, the clustering problem can be formulated as an optimization problem (Selim and Ismail, 1984), which is described by the following equation:

$$P: \text{minimize } z(W, M) = \sum_{i=1}^{n} \sum_{j=1}^{k} w_{ij} d(x_i, c_j)$$

$$\text{subject to } \sum_{j=1}^{k} w_{ij} = 1, \text{for } i = 1, \ldots, n,$$

$$w_{ij} = 0 \text{ or } 1, \text{for } i = 1, \ldots, n, \text{and } j = 1, \ldots, k \tag{1}$$

where $w_{ij} = 1$ implies that object $x_i$ belongs to clusters $G(j)$ and $d(x_i, \mu_j)$ denotes the distance between $x_i$ and $\mu_j$ for $i = 1, \ldots, n$ and $j = 1, \ldots, k$.





The S $k$-means algorithm consists of four steps (**Fig. 1a**), which are similar to that of generic algorithms except for step (*ii*). The steps are described as follows:

- *(i) Initialization.* Initialize $k$ centroid vectors. Although $k$-means has several options for initialization, we apply a randomized scheme to initialize the centroids.
- *(ii) Classification.* Assign an object to its most similar centroid. The S $k$-means algorithm uses the structural similarity
(S-SIM) recognition technique to determine the most similar centroids instead of using distance measures, such as that in generic algorithms.
- *(iii) Centroid calculation*. Update centroid vectors by taking the mean value of the objects belonging to these clusters.
- *(iv) Convergence determination.* The algorithm stops when equilibrium is reached, that is, when there are no object migrations from one cluster to another. If equilibrium is not reached, then the process is repeated from step (*ii*).

S $k$-means is compared with E, M, and C $k$-means ($k$-means using the Euclidean distance, the Manhattan distance, and the Pearson correlation coefficient). E, M, and C $k$-means also follow the same procedure as indicated above except for classification scheme (*ii*), where the respective similarity/distance measures are used to determine the most similar centroids.

**2.2 Structural similarity**

The metrics for the structural similarity (S-SIM) recognition process were first introduced by Wang et al. (2004). It was
developed to better predict the perceived quality of digital television and cinematic pictures. S-SIM is intended to improve the traditional peak signal-to-noise ratio or mean squared error in detecting similarities between "structural" signals, such as images. Intuitively, S-SIM is determined by considering the differences between two input signals (vectors $x, y$) across multiple aspects including "luminance" ($l$), "contrast" ($c$), "and structure" ($s$), which represent the characteristics of human visual perception. "Luminance" measures the similarity in brightness values; "contrast" quantifies the similarity in illumination
variability; and "structure" measures the correlation in spatial inter-dependencies between images (Wang and Bovik,2009). Mathematically, S-SIM is determined as follows:

$$SSIM(x, y) = l(x, y)^{\alpha} \times c(x, y)^{\beta} \times s(x, y)^{\gamma} \qquad (2)$$

where the individual comparison functions are $l(x, y) = \frac{2\mu_x\mu_y + c_1}{\mu_x^2 + \mu_y^2 + c_1}$, $c(x, y) = \frac{2\sigma_x\sigma_y + c_2}{\sigma_x^2 + \sigma_y^2 + c_2}$, and $(x, y) = \frac{\sigma_{xy} + c_3}{\sigma_x\sigma_y + c_3}$. Here, $\mu_x$ and $\mu_y$

represent the average and $\sigma_x$ and $\sigma_y$ represent the standard deviations of vectors $x, y$; and $c_1, c_2,$ and $c_3$ are parameters to





stabilize the division with a weak denominator. For simplification, we set $c_1 = c_2 = c_3 = 0$ and weights $\alpha = \beta = \gamma = 1$ and
reduce the original formula to the following:

$$SSIM(x, y) = \frac{2\mu_x\mu_y\sigma_{xy}}{(\mu_x^2 + \mu_y^2)(\sigma_x^2 + \sigma_y^2)} \qquad (3)$$

S-SIM values range from -1 to 1, where -1 indicates totally dissimilar and 1 indicates totally similar. Wang and Bovik (2009) showed that S-SIM represents a powerful, easy-to-use, and easy-to-understand alternative to traditional distance metrics, such as Euclidean distance, for dealing with spatially and temporally structured data. S-SIM emerged as a "new-generation" similarity metric with an increasing number of applications outside the signal processing field. Moreover, S-SIM has recently
attracted the attention of hydrological and meteorological researchers (e.g., Mo et al., 2014; Han and Szunyogh, 2018; Doan et al., 2021).

## 3. Demonstration tests

S $k$-means is applied to three representative clustering problems. These problems cover various types of input datasets that represent diverse issues, i.e., weather pattern (in terms of two-dimensional pressure data), historical climate change (in terms
of time series), and tropical cyclone tracking data (the hybrid type of data containing both spatial and temporal information) (**Fig. 1b**). The details of these three tests are described below.

- *Weather pattern (WP) clustering*. Group winter weather patterns in Japan. The mean sea level pressure (SLP) was obtained using ERA-Interim reanalysis data (Dee et al., 2011). The data had a horizontal resolution of 0.75°on a regular grid but were re-gridded to an equal-area scalable earth-type grid at a spatial resolution of $200 \times 200$ km. This
160        interpolation/regridding method is commonly applied to high-latitude domains (Gibson et al., 2017). Data collected in winter months, that is, December, January, and February (DJF), for ten years (2005-2014) over the region from 20 – 50 °N and 115 – 165°E were used. The total number of samples used was 902. Each sample had a grid size of 35 pixels $\times$ 35 pixels.

- *Climate change (CC) clustering*. Group temperature-increase time series data collected over 70 years (1951 – 2020)
165        from in situ weather stations run by the Japan Meteorological Agency. A simple data-quality check was implemented. Weather stations that missed (daily basis) observations for more than 10% of the total period of interest were excluded. Therefore, 134 valid weather sites remained (see Fig. 1b CC for the location of weather sites). The annual mean of





each time series was calculated, and the climate change components were determined by subtracting the average of the first 30 years (1951 – 1980) from each value series.

- *Tropical cyclone (TC) tracking clustering.* Group the best TC tracks from 1951 to 2020, which were retrieved from the Japan Regional Specialized Meteorological Center (RSMC) (https://www.jma.go.jp/jma/jma-eng/jma-center/rsmc-hp-pub-eg/besttrack.html). Note that the RMSC provides only the best TC tracks, which have a maximum wind speed of more than $17.2\ m \cdot s^{-1}$, e.g., wind force 8 of the Beaufort scale (Barua, 2019). These data contain the TC classification, maximum sustained wind speed, central pressure, and latitude and longitude of the TC centers with 6-hourly intervals. In this study, only TCs that passed the Japan region, defined as the region between $25 – 45°$N and $126 – 150°$E, were used for the analysis. Hence, the total number of TCs feeding the $k$-means was 863. Because $k$-means clustering requires identical lengths of input vectors, the TC tracks were reconstructed so that they had an equal length of 20 segments by the method proposed by Kim et al. (Kim et al., 2011), which has been applied in several studies (Choi et al., 2012; Kim and Seo, 2016).

As mentioned in the introduction, in addition to S $k$-means, the C, E, and M $k$-means methods that use Pearson correlation coefficients and Euclidean and Manhattan distances for the classification scheme were used for the tests. For this, we performed a total of $3 \times 4 = 12$ simulations. For each simulation, 11 $k$ settings were implemented, that is, $k = 2, 4, 6, …, 20$, and for each $k$, ten runs (randomized initializations) were realized. In summary, a total of $12 \times 11 \times 10 = 1320$ runs (model realizations) were performed for the analysis.

## 4. Evaluation measures

### 4.1 Similarity distributions

The similarity-distributions technique developed by Doan et al. (2021) to evaluate "global" pairwise relationship of input vectors was adopted, and it was named the similarity distribution (S-distribution or S-D) in this study. The S-distribution is a probability density function of pairwise similarities of a vector set. Suppose that $X = \{x_1, …, x_n\}$ is the set of $n$ objects; $s_{ij}$
is the pairwise similarity between two objects, which is defined as $s_{ij} = F(x_i \rightarrow x_j)$; and $F$ is the similarity function, $i = 1, 2, …, n$; $j = 1,2, …, n$. The normalized $s_{ij}$ is defined as $s'_{ij} = (s_{ij} - \min\{s\})/(\max\{s\} - \min\{s\})$. By definition, $s'_{ij}$ ranges from 0 to 1, with the maximum value of 1 indicating perfect similarity (self-similarity) and the minimum value of 0 indicating a lack of similarity (distance to the furthest object), and it is data dependent. As $F$ is a symmetric function, that is,





$F(x_i \rightarrow x_j) = F(x_j \rightarrow x_i)$ for all similarity/distance indices of interest, i.e., S-SIM, COR, ED, and MD, duplicated values

were removed. In addition, results showing self-similarity, that is, $s'_{ij}$ with $i = j$, were removed. Thus, $n(n-1)/2$ values

remained in the final set $S$ of $s'_{ij}$. The S-distribution, or S-D, is defined as the probability density function of the values of $S$.

The S-Ds were then plotted together for comparison. In addition, statistical parameters, such as the mean, standard deviation,

skewness, kurtosis, and Shannon entropy, were calculated to further diagnose the characteristics of the datasets of interest.

### 4.2 "General" silhouette analysis

In this study, we focused on *k*-means clustering as an unsupervised machine learning method. Thus, we assume the absence

of "ground truth," or predefined cluster labels of a given dataset in which the goodness of the cluster outcomes is defined.

Therefore, an internal validation approach was adopted. In the internal validation, clustering is only compared with the result

itself (Hassani and Seidl, 2017) based on the criterion that the clustering methods must group objects to optimize the

homogeneity within a cluster and maximize the difference among clusters. Many internal indices have been used for clustering

evaluation. However, a problem with conventional indices when dealing with non-distance metrics is that these indices are

built on Cartesian geometric algebra, which is not the case with non-distance metrics.

This study uses the general silhouette analysis method to validate the algorithms. Silhouette analysis is a comprehensive

analysis of the interpretation and validation of cluster methods. This technique offers a concise graphical representation of

how well each object has been classified (Rousseeuw, 1987). The silhouette value is a measure of how coherent an object is

with its cluster versus how it is separated from other clusters. The general silhouette analysis is the generalized form of the

silhouette analysis that can applicable also for non-distance metrices. This concept was firstly used for the evaluation of self-

organizing maps by Doan et al. (2021). Mathematically, the general silhouette coefficient ($GSC$) for a given object is defined

as follows:

$$GSC = \frac{b - a}{\max\{a, b\}} \qquad (4)$$

where $a \; and \; b$ are the mean intracluster distance and mean distance to the nearest cluster, respectively. Note that the distance

here is the "general" distance and not the Euclidean distance, which was originally defined in the study by Rousseeuw (1987).

The general distance is the reversed normalized similarity (i.e., $-s'_{ij}$) defined in subsection 4.1, which is why here we call it

the general silhouette coefficient.





The $GSC$ values ranged from −1 to +1. A higher value indicates the goodness of the cluster assignments, that is, the object is coherent with its cluster and well separated from neighboring clusters. The clustering configuration is appropriate if most objects have high scores. In contrast, if many objects have low or negative values, then the clustering configuration performs poorly. A $GSC$ of zero indicates that the object is on or very close to the border of two neighboring clusters, and a negative $GSC$ indicates that the object may have been assigned the wrong cluster label.

### 4.3 Clustering uncertainty evaluation

Evaluating the variability/uncertainty inherent in a clustering algorithm is challenging owing to the unique nature of the clustering outcome. On one hand, compared with other statistical random variables, it is difficult to define the statistical mean, standard deviation, or range between quantiles of a given ensemble of clustering realizations. Therefore, it is impossible to use statistical measurements to represent associated variability/uncertainty. On the other hand, applying a clustering algorithm as an unsupervised learning method assumes the absence of "ground truth" or "absolute reference" so that the variance from that could be easily defined.

Herein, we propose a framework for the representation/evaluation of the variability/uncertainty of the clustering problem. The framework is based on a pairwise comparison of clustering realizations using a quantified index called the clustering uncertainty degree (CUD). The CUD is based on the mutual information concept; in more detail, it is a recently developed adjusted mutual information index. In information theory, mutual information from two random variables is used to quantify the "amount of information" obtained for one random variable by observing another random variable. The concept of mutual information is intimately linked to the entropy concept of a random variable, which is a fundamental notion in information theory that quantifies the expected "amount of information" held in this variable. Recently, mutual information has been applied to evaluate the agreement between two clustering assignments. To do so, the mathematical formula for mutual information $I(U,V)$ between two clustering realizations (label assignments of $N$ objects) $U$ and $V$ is defined as follows:

$$I(U,V) = H(U) + H(V) - H(U,V) \tag{5}$$

where $H(U)$ $and$ $H(V)$ are the entropies of each realization and $H(U,V)$ is the joint entropy of the two. Entropies of clustering realizations are defined as the amount of uncertainty for partition sets.





$$H(U) = - \sum_{i=1}^{|U|} P(i)\log(P(i)) \tag{6}$$

$$H(V) = - \sum_{j=1}^{|V|} P'(j)\log(P'(j)) \tag{7}$$

where $P(i) = a_i/N$ and $a_i = |U_i|$ is the probability that an object pickup at random from $U$ falls into class $U_i$. Similarly, for $V$, $P'(j) = b_j/N$, where $b_j = |V_i|$ is the probability of an object from $V$ falling into class $V_j$.

$$H(U,V) = - \sum_{i=1}^{|U|}\sum_{j=1}^{|V|} P(i,j)\log(P(i,j)) \tag{8}$$

where $P(i,j) = |U_i \cap V_j|/N$ is probability that an object pickup at random falls into both class $U_i$ and $V_j$.

By definition, mutual information ranges from 0 to 1. A value of 1 indicates perfect agreement (equality) between the two
clustering realizations, while values close to zero indicate that the two label assignments are largely independent. The drawback of mutual information is the possibility of weakness. Vinh et al. (2009) derived the expected mutual information and proposed the concept of adjusted mutual information that can defend against chance (Vinh and Epps, 2009; Vinh et al., 2010; Romano et al., 2016). Thus, random (uniform) label assignments have an adjusted mutual information score close to 0.0 for any number of clusters and objects (which is not the case for raw mutual information).

$$E[I(U,V)] \tag{9}$$

$$= \sum_{i=1}^{|U|}\sum_{j=1}^{|V|}\sum_{n_{ij}=(a_i+b_j-N)^+}^{\min(a_i,b_j)} \frac{n_{ij}}{N}\log\left(\frac{Nn_{ij}}{a_ib_j}\right)\frac{a_i!\,b_j!\,(N-a_i)!\,(N-b_j)!}{N!\,n_{ij}!\,(a_i-n_{ij})!\,(b_j-n_{ij})!\,(N-a_i-b_j+n_{ij})!}$$

$$I'(U,V) = \frac{I(U,V) - E[I(U,V)]}{mean\{U(U), H(V)\} - E[I(U,V)]} \tag{10}$$



The core concept underlying the CUD, i.e., clustering uncertainty degree, is defined as follows:

$$CUD(U,V) = 1 - I'(U,V) \qquad\qquad ((11)$$

By definition, CUD is a representation of pairwise disconsensus of clustering realizations. The CUD ranges from $0 - 1$. A value of 1 indicates the greatest disconsensus or highest uncertainty between $U$ and $V$, while a value of 0 indicates perfect consensus or no uncertainty. The connectivity matrix of pairwise CUDs is defined as a $M \times M$ matrix and CUD values for a pair of clustering realizations, where $M$ is the number of clustering realizations. The connectivity matrix naturally serves as a

visualization tool to assess the general uncertainty of the clustering system. Other visualization tools are also used to visualize the CUD, including a heatmap and a chord diagram (Holten, 2006). Heatmaps work like a connectivity matrix but in a more visualized form. A chord diagram is a useful graphical method for demonstrating the interrelationships between the data in a matrix. The data are plotted radially around a circle. The relationships between data points are usually drawn as arcs that connect the data.

**5. Results and discussion**

**5.1 S-distributions**

Before analyzing the *k*-means clustering results, we diagnosed the nature of the input data using S-distributions (or S-D). S-Ds provide "global" insights into how data vectors are related to each other in four S-SIM, COR, ED, and MD topological spaces. The results, which are shown in **Figure 2**, demonstrate an apparent difference in the shape of the S-Ds. Notably, the

S-Ds for ED and MD appeared more symmetrical than those for S-SIM and COR across the three types of input data, that is, WP, CC, and TC. For S-SIM and COR, S-Ds tended to be more tailed (both sides), with skewness over the left tail. Quantitively, the standard deviation of S-Ds for S-SIM and COR tended to be higher $(0.13 - 0.20)$ than those for ED and MD (approximately $0.11 - 0.13$) (Table 1), despite an exception for ED in the TC simulation. The consistent skew-over-left of S-SIM and COR indicates that those tend to project "hierarchical affinity" of input vectors, meaning that a given vector tends to be closer to a

certain group of peers and relatively far from another group located at the opposite end of similarity spectrum. These results demonstrate that the discrimination ability of S-SIM and COR is higher than that of traditional distance metrics, such as ED or MD.





## 5.2 Clustering results

As explained in Section 3, three demonstration problems, WP, CC, and TC, were conducted with different $k$ configurations
and centroid initializations, with a total number of runs of 1320. The results are shown partly in **Figure 3, 4, and 5**. Recall that
this study addresses the algorithm aspects (attempting to seek general insight into the system's performance regardless of
problems). We do not intend to physically interpret the specific clustering outcomes, although some phenomenal explanations
are provided in the manuscript.

The clustering results are partly visualized and shown together with quantified silhouette scores in **Figures 3, 4**, and **5** for WP,
CC, and TC, respectively. Note that only the configuration $k = 4$ for the first initialization (R0) is shown. For other
initializations ( R1–9), see the Supplementary Material. First, we examined the $k$-means-detected weather patterns over the
Japan region (**Fig. 3**). During December, January, and February (DJF), the weather in Japan is dominated by a winter-type
pattern. The winter type is characterized by the Siberian High, which develops over the Eurasian continent, and the Aleutian
Low, which develops over the northern North Pacific. The location of the high and low winds resulted in prevailing
northwesterly winds. The wind blows cold air from Siberia to Japan and causes heavy snowfall on the western coast and sunny
weather on the Pacific side of the country. This winter-type pattern is clearly captured by all types of $k$-means, that is, C2 for
S, C4 for C, C3 for E, and C4 for M $k$-means (**Fig. 3**). An interesting result was revealed by the silhouette analysis. The S $k$-
means method generated dominant C2 over other types regarding its frequency (the thickness of each cluster label in the
silhouette diagram indicates the number of members in the cluster). This result is consistent with prior knowledge of the
weather patterns over the region (https://www.data.jma.go.jp/gmd/cpd/longfcst/en/tourist_japan.html). Moreover, the S $k$-
means algorithm showed consistently higher silhouette scores than the other algorithms for all $k = 2, 4, ..., 20$ settings (**Fig.
6a**). The highest score for S $k$-means was followed by that for C $k$-means. E and M $k$-means consistently had lower scores than
S and C $k$-means.

Regarding the CC experiment, the time-series results were visualized with reference to the real geographical locations of the
weather stations to support the interpretation (**Fig. 4**). CC is aimed at clustering the 70-year time series of temperature
anomalies (from the 1950 – 1979 average). Overall, the analysis shows that although a temperature increase is seen over all
stations, this trend is not geographically uniform. These regional differences were well captured by $k$-means clustering. For
example, the northern part (Hokkaido) is consistently separated from other regions in terms of temperature warming. This
finding implies faster warming in the north than in other regions. Although this is not the main concern of this study, such a
result highlights the usefulness of $k$-means to detect regional differences, which is useful for building detailed appropriate





climate change actions. Regarding clustering quality, a similar superiority of S and C *k*-means was confirmed. Similar to WP, the S and C *k*-means had relatively higher silhouette scores for the CC data compared with E and M *k*-means (**Fig. 6b**).

In addition, the TC experiment aimed to determine how *k*-means works with hybrid spatiotemporal data. Similar to the above experiments, for the WP and CC data, the S and C *k*-means methods outperformed the E and M *k*-means methods, as reflected by the higher silhouette scores shown in **Figure 5** and **Figure 6c**. In addition, the results of all clustering methods were consistent, with the silhouette scores decreasing and being stable when *k* was equal to or larger than four. This suggests that for clustering the TC track, the optimal number of statistically distinctive cluster pattern pairs for *k* is four. **Figure 5** shows the four main patterns of the TC track determined using the four clustering methods. Although there are some differences in the TC track between the results of the tested clustering methods, such as genesis and depression points, all determined patterns are characterized mainly by curved trajectories. These averaged patterns could be divided into two groups: i) not crossing and ii) crossing mainland Japan. Overall, the number of TCs in group i) was higher than that in group ii), with these tracks characterized by TCs containing both straight and re-curving TC trajectories forming to the east of 140° E (e.g., clusters 2 and 4 of S *k*-means in **Fig. 5a**). For group ii), the averaged patterns show the TC track passing through the central area of Japan (e.g., clusters 1 and 3 of S *k*-means in **Fig. 5a**)

Consistently, a higher performance of S *k*-means was observed throughout the ensemble of tests, *k* settings, and initializations. The performance of S *k*-means is sometimes completed using C *k*-means. The two S and C *k*-means algorithms outperform the distance-metrics-based E and M *k*-means algorithms. However, it is worth noting that these results were obtained from the silhouette analysis. Additional evaluation approaches might be needed to generalize the conclusions, although this could be challenging because most objective clustering evaluations have been developed on the Cartesian geometric algebra assumption (that could work for distance metrics but might not work for non-distance measures). Therefore, to address this issue, it is necessary to develop new evaluation approaches beyond the distance paradigm. Another difficulty in *k*-means evaluation is that, similar to other clustering techniques, *k*-means is classified as an unsupervised machine learning method. This means that there is an absence of a single "ground truth," referring to the definition of the goodness of the clustering result. We also suggest diversifying the clustering problems with different types of input data, or for different geographical areas, to obtain a more comprehensive picture of the proposed algorithm and its advantages and disadvantages. We also suggest linking the cluster results for prediction purposes, for that the "goodness" of the clustering algorithm can be determined in practice. Such questions could be useful for future research.





Computational cost is also an important factor, especially in a practical sense. We measured the computational cost of obtaining the results, which are shown in **Figure 7**. Overall, S $k$-means and C $k$-means require more time to complete the same task than

E and M $k$-means. Roughly, S $k$-means required $5 - 6$ times more computational time than E $k$-means. C $k$-means was comparable to S $k$-means. M $k$-means required less computational time than E $k$-means. Such a tradeoff between higher performance and computational cost should be considered when selecting an algorithm. However, the general computational cost was a big problem; for example, in this study, the time to finish one run was mostly less than a minute, which was very small compared to the numerical weather prediction or climate simulation. In addition, the computational issue can be solved

by drastically improving the computational ability or using a parallel computational approach.

### 5.3 Uncertainty evaluation

The results of the proposed clustering uncertainty evaluation framework (CUEF) is discussed here. The clustering uncertainty degree (CUD) is shown in **Figure 8** (for $k = 4$ and run R0; the collective results are shown in **Figure 10**). As explained in Section 4, two visualization tools, that is, heat maps and chord diagrams, were proposed for diagnosing clustering uncertainty.

Accordingly, pairwise differences and overall variability among clustering outcomes are comprehended in both quantified and qualified manners. For example, from **Figure 8a** (WP), the CUD values for S relative to C, E, and M $k$-means are 0.67, 0.75, and 0.77, respectively, according to the heatmap. Note that the maximum value (1.0) indicates the absolute disagreement between two clustering assignments and the minimum value (0.0) indicates the absolute consensus between the two. However, based on the same results, the chord diagram demonstrates the pairwise relationship in a more qualified manner. From the

diagram, one can easily determine which algorithms (S, C, M, or E) have less consensus with another (wide arc length on the circle means less consensus), and vice versa. Notably, the results show that E and M $k$-means showed a higher consensus with each other. Particularly in the CC and TC experiments, S $k$-means showed less uncertainty/high consensus relative to E and M compared with C $k$-means.

In addition to the algorithm-wise uncertainties, this study evaluated the initialization-wise uncertainties. The $k$-mean results

are sensitive to how centroids are initialized. One might want to apply this proposed framework to evaluate initialization-induced clustering uncertainty. Here, heat maps in which pairwise CUDs between runs (i.e., R0 – R9 for each simulation) are shown in **Figure 9** for the WP, CC, and TC experiments, and four $k$-means algorithms. Note that we do not show a chord diagram here because such a diagram is not relevant if the number of comparing elements is sufficiently small and have actual meaning that must be discussed. This is not the case when R0 – R9 are compared. The results demonstrate that there is less

uncertainty regarding initialization than that owing to the selection of $k$-means algorithms (**Fig. 9 and 10**). In particular, the





initialization-wise CUDs are lower than the algorithm-wise CUD for WP and TC. Meanwhile, in CC, the initialization-wise and algorithm-wise CUDs do not show apparent differences except for $k < 6$ (**Fig. 10**).

The above results demonstrate the effectiveness of CUEF with the core concept of CUDs used within visualization frameworks, such as heatmaps or chord diagrams, to quantitively represent/evaluate the uncertainty inherent in clustering outcomes.
Heatmaps and chord diagrams are useful in offering intuitive and general comprehension of uncertainty and consensus among the outcomes. CUEF can be used to evaluate algorithm-wise and initialization-wise uncertainties, or both. Note that CUEF is not limited to $k$-means clustering and can also be applied to other clustering algorithms. In addition, note that most clustering approaches use a type of randomized initialization to start the learning procedures, and CUEF can be used to evaluate the intrinsic uncertainty of these algorithms. As the first study to address this issue, we believe that CUEF can constitute a new
standard for addressing uncertainty issues when performing data clustering in (but not limited to) climate science.

An additional important benefit of CUEF is that it can be used to answer the fundamental question of how meaningful it is to apply a clustering solution to a given problem. For example, in **Figure 8 and 9**, which compare WP, CC, and TC, the CUDs for WP are usually higher than those for CC and TC (higher CUD values are intuitively identified by a lighter color). This finding indicates that WP can yield more random clustering outcomes regardless of the algorithm used or initialization;
therefore, the problem lies in the data itself and not the clustering procedure. This makes sense because different data have different topologies, which can make them unsuitable or even invalid for a clustering solution. The question of whether it is valid or meaningful to apply a clustering solution to a dataset is more important than how to find the best method of clustering. Although this issue is fundamentally important, to the authors' best knowledge, no studies have addressed this question or proposed a solution, at least among the climate sciences. We believe that CUEF can be used to answer the question of whether
it is meaningful to apply clustering to a given dataset. In this paper, although we proposed three demonstration problems for comparison, more diverse problem settings should be implemented in the future to test the effectiveness of CUEF.

## 6. Summary and remarks

This study proposes (i) a novel $k$-means algorithm primarily for mining climate data and (ii) a clustering-uncertainty evaluation framework. The novel $k$-means algorithm, called S $k$-means, is characterized by its ability to deal with inherent spatiotemporal
"structuredness" in climate data. In detail, S $k$-means incorporates the recent innovation in signal recognition regarding structural similarity into the classification scheme, which has been developed on the distance metrics paradigm.



The performance of S $k$-means was evaluated against other $k$-means using other distance/similarity metrics, namely, the Pearson correlation coefficient, Euclidean, and Manhattan distance (C, E, and M $k$-means, respectively) for multiple demonstration tests: clustering weather patterns (spatial-related data), historical climate change (time series) for long-term recorded weather station data, and best tracks of tropical cyclones (spatiotemporal hybrid). Multiple $k$ settings ($k = 2, 4, \ldots, 20$) for each $k$ and an ensemble of 10 randomized initializations were implemented, resulting in a total of 1320 runs produced to generate robust results.

The quantitative approaches similarity distribution (S-D) and general silhouette analysis were used to evaluate the performance of the algorithm. S-D diagrams were used to diagnose the topological relationship of input datasets after projection onto different distance/similarity spaces, and they showed that structural similarity matrix groups are likely to have a higher ability to discriminate the data (characteristics that might be useful for clustering) than conventional distance metrics. Regarding the clustering results, the general silhouette analysis showed consistently higher scores for S and C $k$-means compared with E and M $k$-means. The superiority of S $k$-means clustering may be achieved by $k$-means clustering. However, S $k$-means consistently outperformed E- and M-$k$-means. It is worth noting that a higher clustering performance often requires more computational resources, and S $k$-means requires five to six times more computational time than E $k$-means.

The higher performance of S $k$-means indicates its promise as a new standard for climate-data clustering/mining, which is a rising research field within the big data context. Nevertheless, certain issues must be noted when interpreting the results of this study. First, $k$-means clustering is assumed to be an unsupervised data-mining method. In other words, there is an assumption of no ground-truth labeling information, which is the reference used to define the goodness of the result. Instead, the goodness of the algorithm is evaluated based on an objective calculus approach using the general silhouette analysis/score. This score is free from the Cartesian geometry assumption, thus allowing the algorithms to be compared with non-distance metrics. Nevertheless, it is suggested that more non-Cartesian-geometry evaluation scores be developed and used to evaluate non-distance clustering algorithms, particularly S $k$-means, in the future.

Finally, another important contribution of this study is that we built a framework for clustering uncertainty evaluation for the first time, and it is primarily applicable to climate research. The evaluation framework is built on the mutual information concept, which was recently developed to quantify divergence of one clustering from its "ground truth". This is the first time this concept has been adapted for clustering uncertainty evaluations in the form of the "clustering uncertainty degree" (CUD) concept. The CUD can measure pairwise discrepancies among clusters, and the collective CUDs can provide an overall picture of the variability/uncertainty of cluster algorithms. Naturally, the CUD can be used to evaluate whether a given problem (input



vectors) is preferable for clustering. In other words, if the cluster algorithm provides higher uncertainty in its outcomes, then it is not appropriate for use, and vice versa. Note that the question of whether it is valid or meaningful to apply a clustering solution to a dataset is more important than determining the best method to apply. For example, for what shown in this study, the WP problem caused more uncertainty in clustering than the CC and TC problems. Thus, it is questioned about the "meaningfulness" of the clustering application for WP compared with CC and TC. We expect this clustering-uncertainty-

evaluation framework will change the conventional agenda of data clustering by adding a procedure to evaluate its application's meaningfulness/effectiveness for a given data.

**Code availability**

The exact version of the model used to produce the results in this study and the input data and scripts used to run the model and plot all the simulations presented in this paper have been archived on Github (https://github.com/doan-van/S-k-means) or

Zenodo (https://zenodo.org/record/6976609).

**Author contribution**

Quang-Van Doan designed the model and developed the model code. Toshiyuki Amagasa, Thanh-Ha Pham, Takuto Sato, Fei Chen and Hiroyuki Kusaka helped to design the test experiments. Thanh-Ha Pham, and Takuto Sato helped to analyze the results. Quang-Van Doan prepared the manuscript with contributions from all co-authors.

**Competing interests**

The authors declare that they have no known competing interests or personal relationships that could have appeared to influence the work reported in this paper.

**Acknowledgements**

The first author, Quang-Van Doan, is sponsored by JSPS KAKENHI Grant Nos. 20K13258, JSPS KAKENHI Grant Nos.

19H01155, and JSPS KAKENHI Grant Nos. 21K03656.



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

**List of tables**

**Table 1. Statistical metrices of S-distributions for three demonstration input datasets, i.e., weather pattern (WP), climate change (CC), and tropical cyclone (TC). The different distance/similarity measures are structural similarity (S-SIM), the Pearson correlation coefficient (COR), Euclidean distance (ED) and Manhattan distance (MD). Statistical measures include the mean (Mean), standard deviation (STD), skewness (SKEW), kurtosis (KUR) and Shannon entropy (ENTROPY)**

|  | WP | | | | CC | | | | TC | | | |
|---|---|---|---|---|---|---|---|---|---|---|---|---|
|  | S-SIM | COR | ED | MD | S-SIM | COR | ED | MD | S-SIM | COR | ED | MD |
| *Mean* | 0.68 | 0.71 | 0.67 | 0.68 | 0.71 | 0.81 | 0.66 | 0.65 | 0.81 | 0.87 | 0.65 | 0.69 |
| *STD* | 0.18 | 0.19 | 0.11 | 0.11 | 0.20 | 0.13 | 0.12 | 0.13 | 0.14 | 0.11 | 0.15 | 0.13 |
| *SKEW* | -0.66 | -0.81 | -0.73 | -0.74 | -1.08 | -1.25 | -0.65 | -0.67 | -1.10 | -1.67 | -0.46 | -0.59 |
| *KUR* | -0.18 | 0.00 | 0.58 | 0.64 | 0.97 | 1.79 | 0.59 | 0.58 | 1.15 | 3.31 | -0.32 | 0.03 |
| *ENTROPY* | 2.83 | 2.79 | 2.19 | 2.16 | 2.83 | 2.29 | 2.32 | 2.36 | 2.30 | 1.80 | 2.57 | 2.45 |






**List of figures**

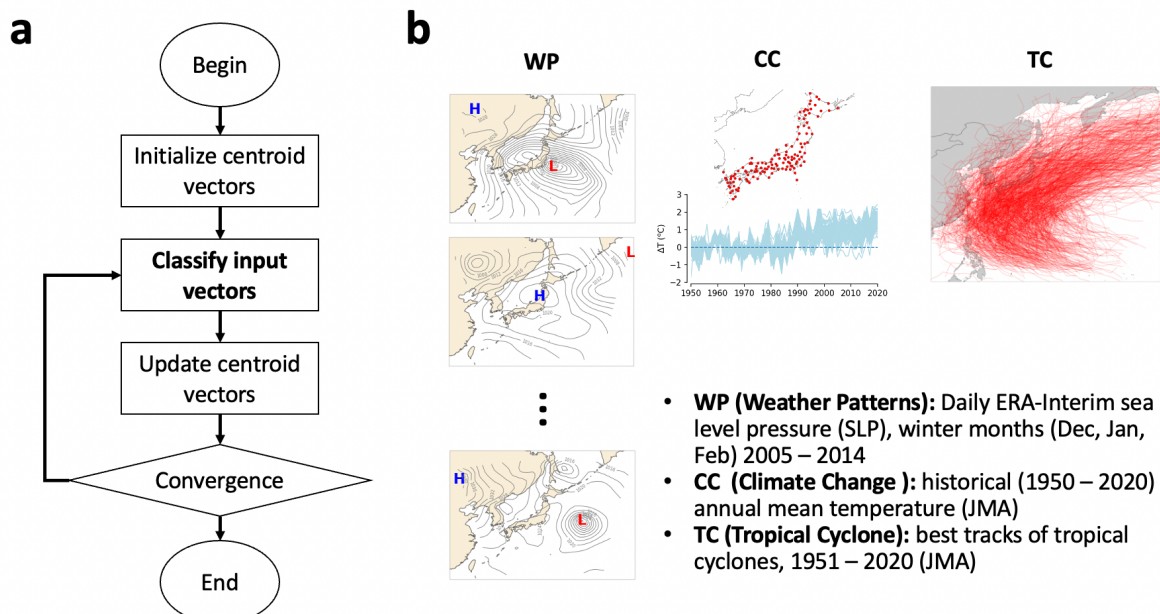

Fig. 1 Illustration of the *k*-means clustering algorithm (a) and three demonstration experiments (b). Demonstration experiments
include clustering weather patterns (WPs) in terms of daily ERA-Interim sea level pressure (SLP) during winter months (December,
January, and February) for ten years 2005 – 2014 over the Japan region; clustering climate change (CC) in terms of historical (1951
– 2020) annual mean temperature collected from in situ weather stations in Japan; and clustering best tracks of tropical cyclones
that passed the Northwest Pacific region from 1951 – 2020. Data were obtained from the JMA

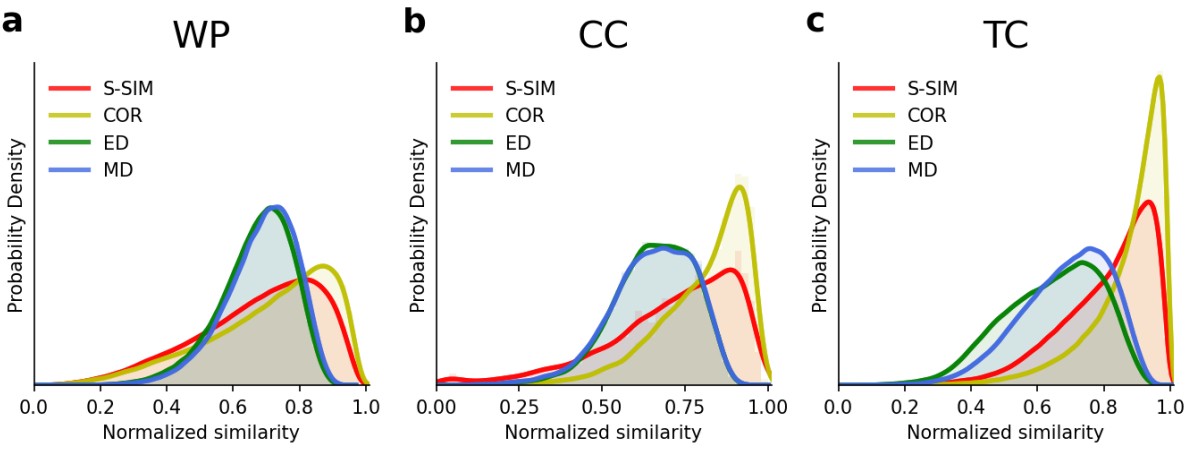


**Fig. 2 Comparison of the S-distributions of normalized pairwise similarity using the structural similarity (S-SIM), the Pearson correlation coefficient (COR) the Euclidean distance (ED) and the Manhattan distance (MD) for three demonstration experiments: WP, CC, and TC. With a population size of N, $\frac{N(N-1)}{2}$ values of pairwise similarity are observed because S-SIM, COR, ED and MD are symmetric measures and self-similarity is excluded. Values are normalized from 0 to 1. The maximum similarity is 1, which corresponds to completely similar, and the minimum similarity is 0, which corresponds to the lowest pairwise similarity.**






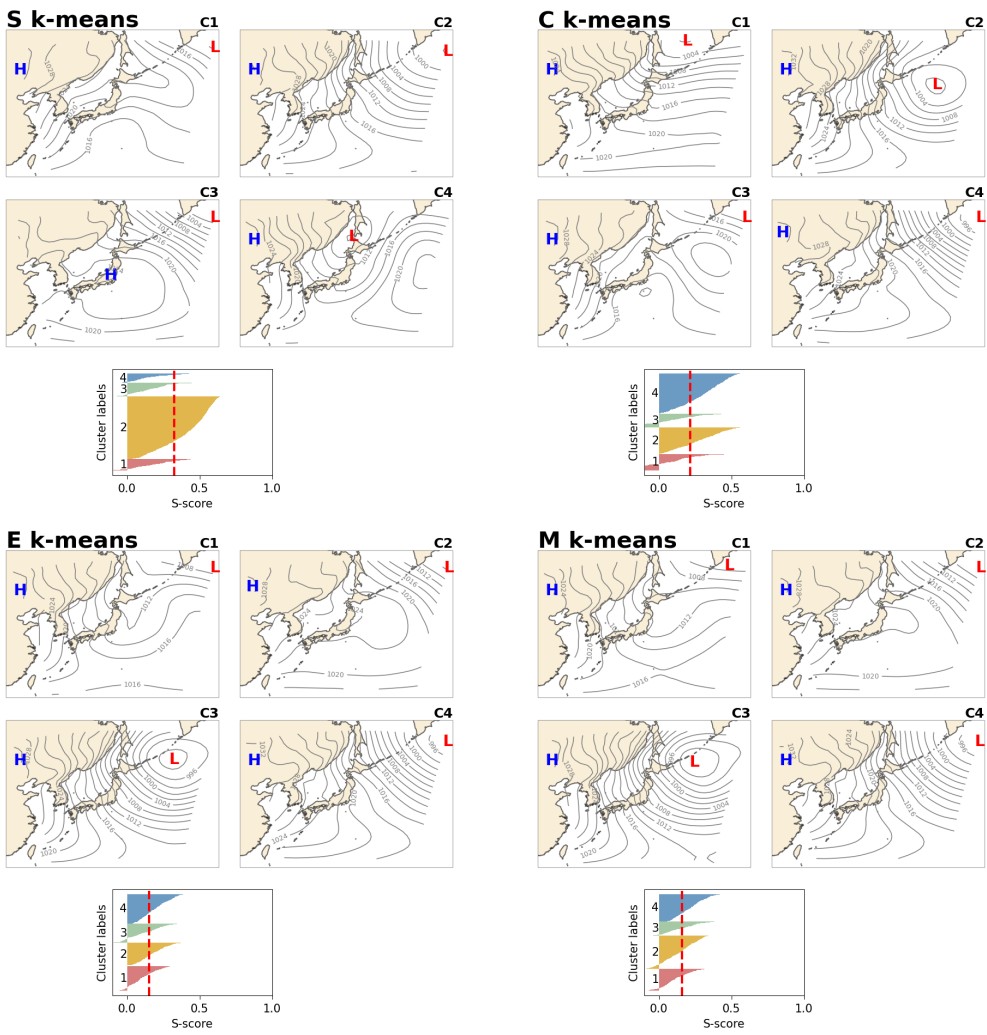

**Fig. 3 Result for the WP experiment. The winter SLP pattern revealed by S, C, E, and M *k*-means with *k* = 4. "H" indicates the location of the high, and "L" indicates the location of the low. General silhouette analysis results are shown below the maps, where the x-axis indicates the score and the y-axis presents the labels of clusters numbered 1 − 4. Input data are ERA-Interim SLP data, which were re-gridded to Cartesian coordinates with a resolution of 200 x 200 km and grid size of 35 x 35. Daily data for December, January, and February collected over ten year 2005 − 2014 were used.**



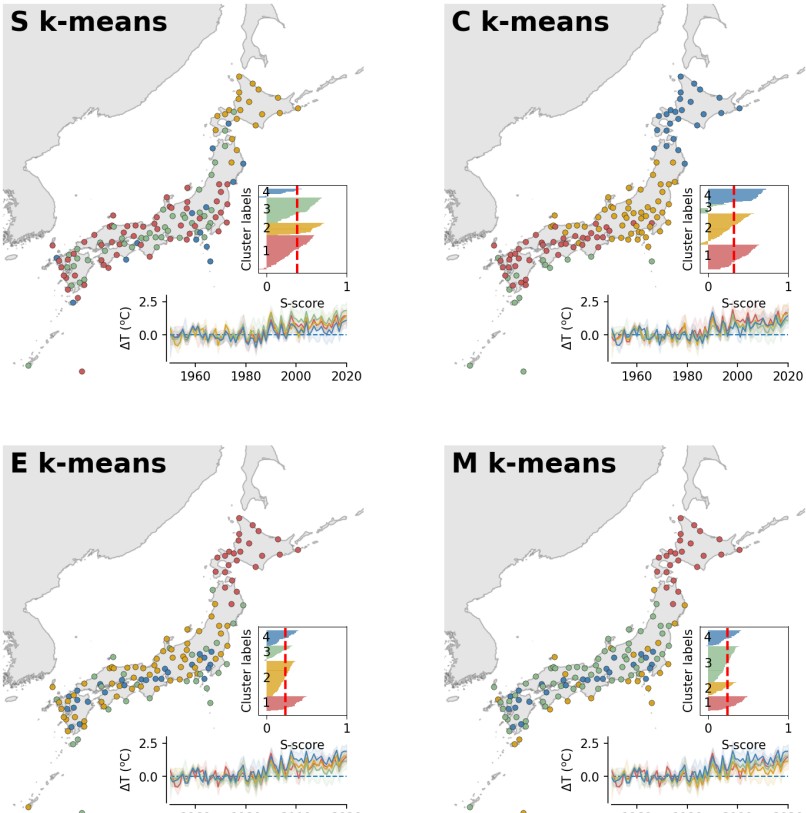

**Fig. 4 Result for the CC experiment for clustering of climate change (temperature increase) time series over 134 weather stations over the entirety of Japan. Patterns were revealed by S, C, E, and M *k*-means, with $k = 4$. Input data correspond to annual mean data collected over 70 years from 1951 – 2020 (subtracted by the mean of the first 30 years) and observed temperature achieved at**

**in situ weather stations (dots in map) operated by the JMA. Time series of centroids and input vectors are shown in below panels together with general silhouette analysis results, where the x-axis indicates the score (S-score) and the y-axis presents the labels of clusters numbered 1 – 4.**





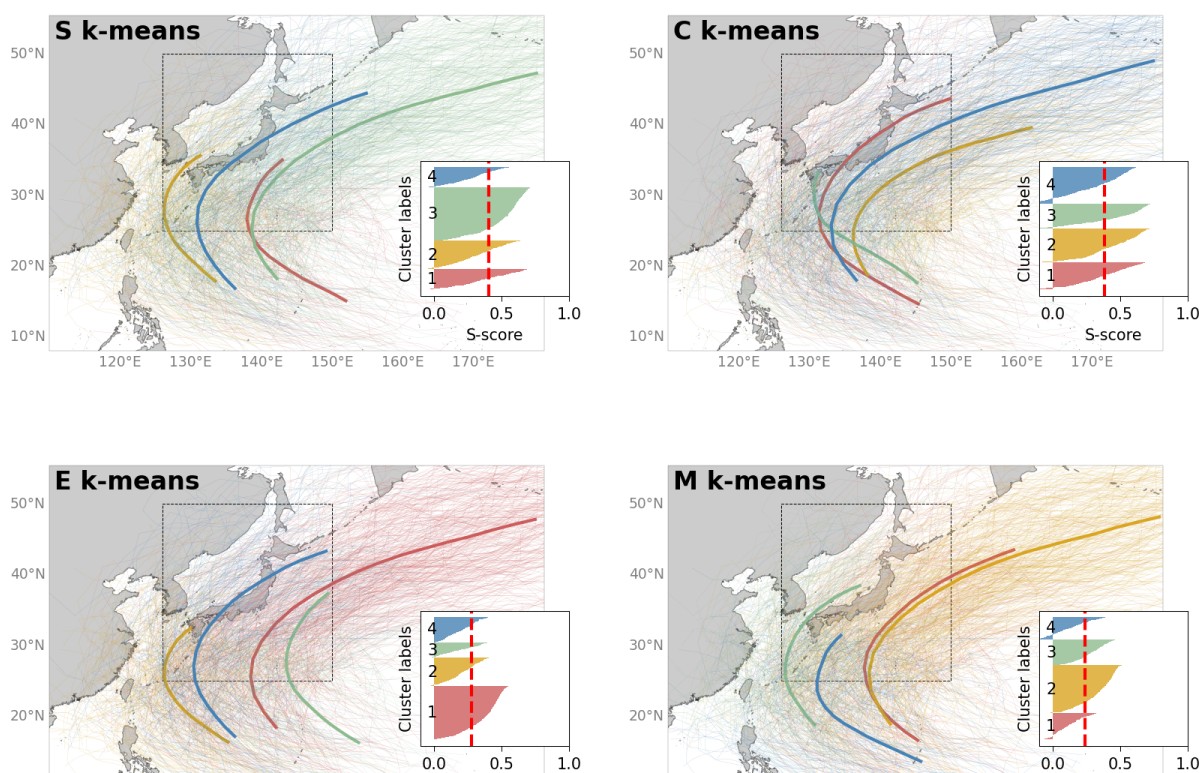

**Fig. 5 Results of the TC experiment for clustering tropical cyclone paths. The pattern was revealed by S, C, E, and M *k*-means, with**
***k* = 4. Input data are the best TC tracks obtained by the JMA from 1951 − 2020. Only TCs that passed the dashed box in the map are used to feed the *k*-means. Thus, a total of 863 TC tracking data points are used. The left side of each panel show the general silhouette analysis results, where the x-axis indicates the score (S-score) and y-axis presents the labels of clusters numbered 1 − 4. The centroid TC path is illustrated by the bold line, and the color is consistent with that in the silhouette diagram.**



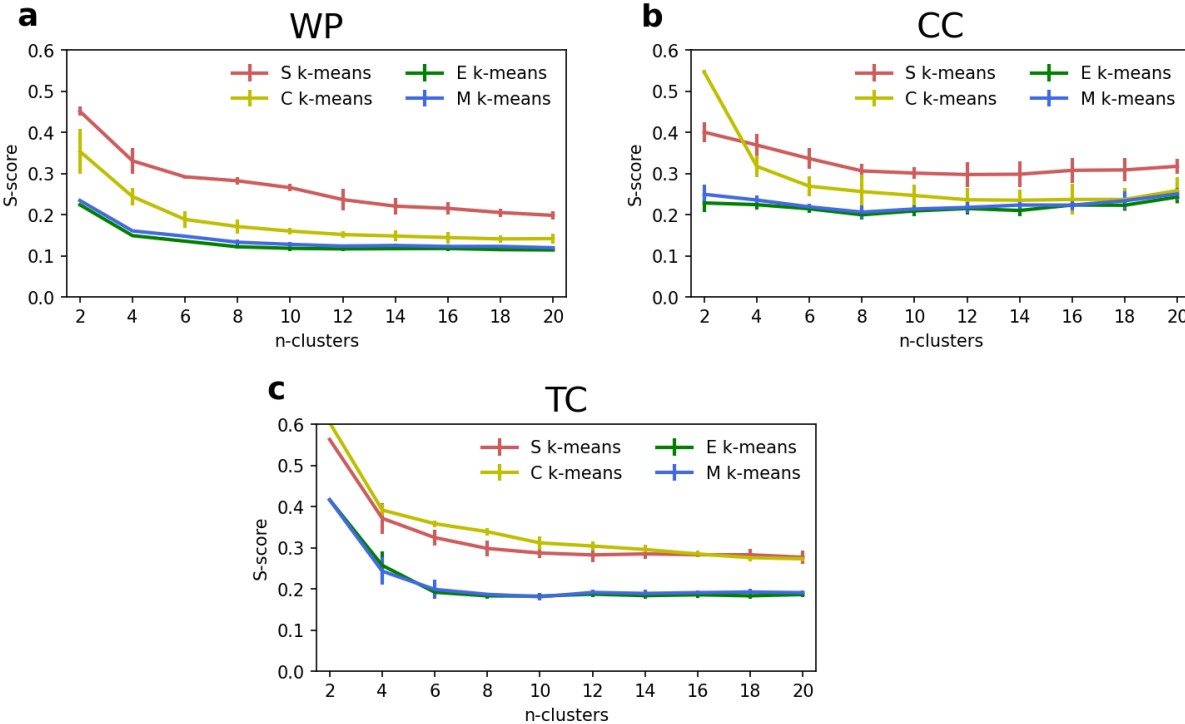

**Fig. 6 Comparison of the average silhouette score (S-score) of S, C, E, and M *k*-means for k = 2, 4, …, 20 for three demonstration experiments: WP (a), CC (b) and TC (c). The uncertainty range in each line indicates the standard deviations of the scores among 10 runs with randomized initializations.**



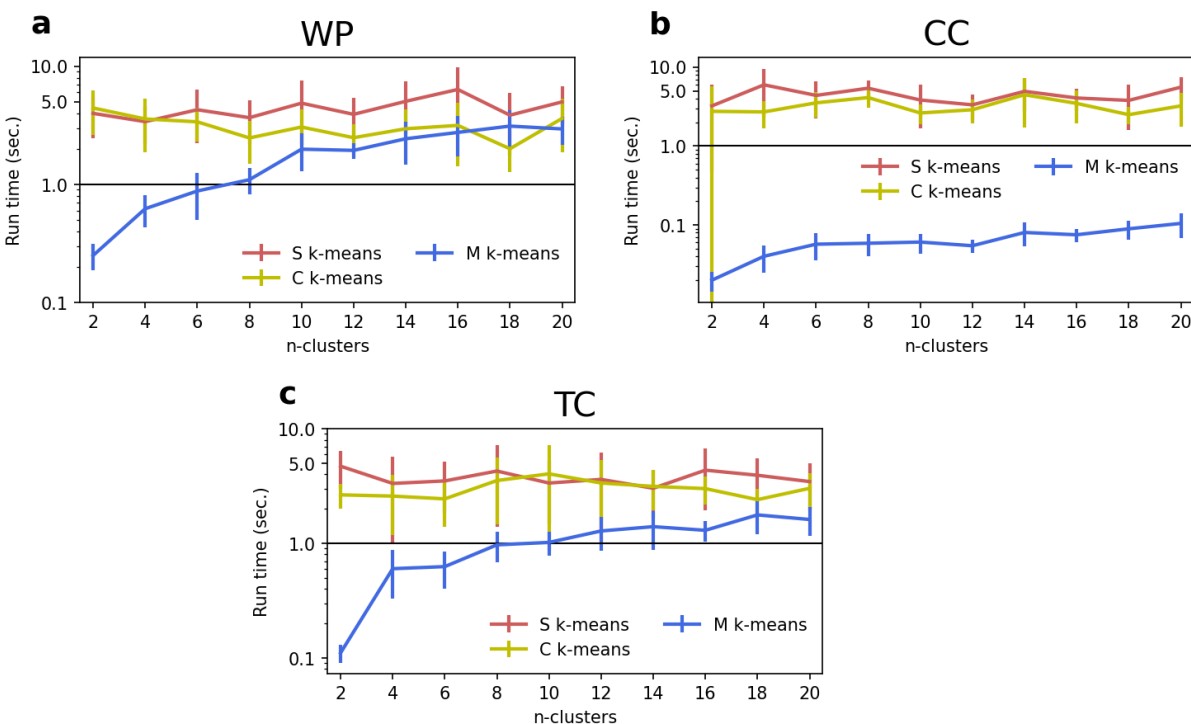

**Fig. 7 Comparison of the run time (in sec) of S, C, E, and M *k*-means for k = 2, 4, …, 20 for three demonstration experiments: WP**
**(a), CC (b) and TC (c). The uncertainty range in each line indicates the standard deviation of the scores among 10 runs with**
**randomized initializations. Note that the y axis is logarithmically rescaled.**



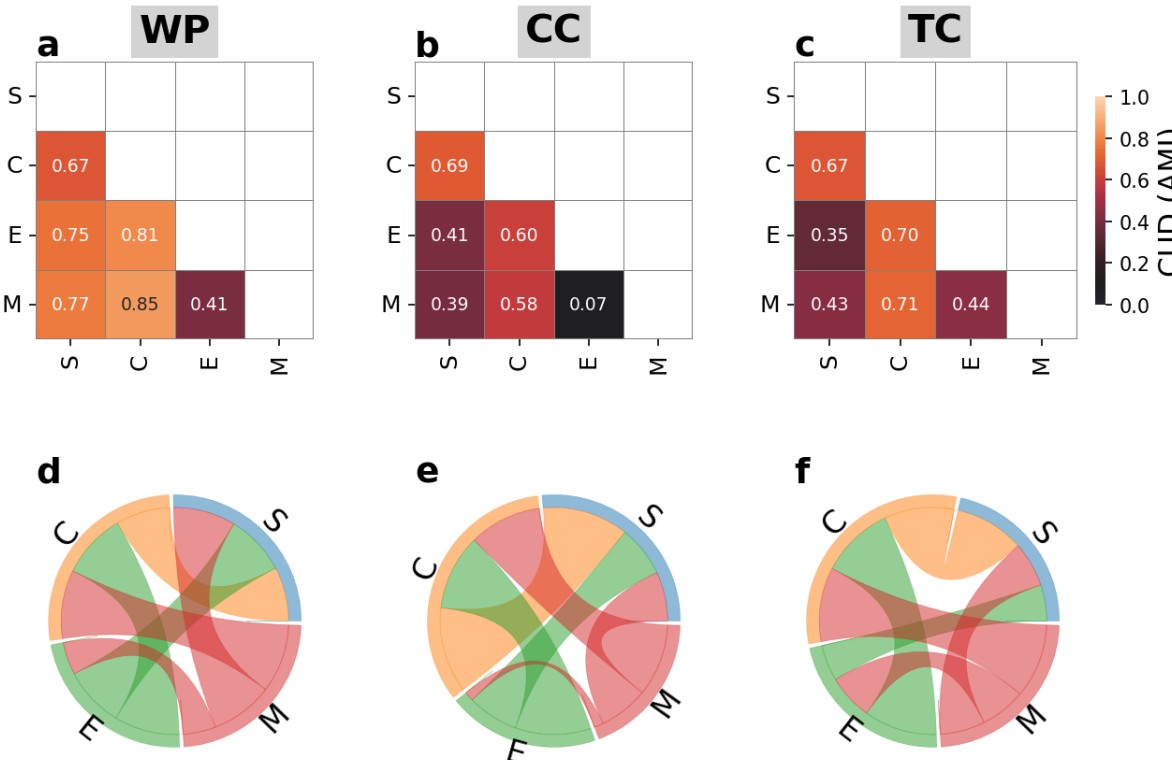

**Fig. 8 Clustering uncertainty degree (CUD) based on adjusted mutual information (AMI) between clustering results from different *k*-means algorithms, i.e., S, C, E, and M *k*-means, for different demo experiments: WP, CC, and TC. (a, b, c) CUD in heatmaps, and (d, e, f) visualization of the interconnection using the chord diagrams. Note that the results are from the configuration with *k* = 4 and the first initialization run.**









**Fig. 9 Clustering uncertainty degree (CUD) based on adjusted mutual information (AMI) between the clustering results from different runs (10 runs indicated by R0, R1, …, R9) of different *k*-means algorithms, i.e., S, C, E, and M *k*-means (rows), for different demo experiments: WP, CC, and TC (columns). Note that the results are from the configuration with *k* = 4 and the first initialization run.**


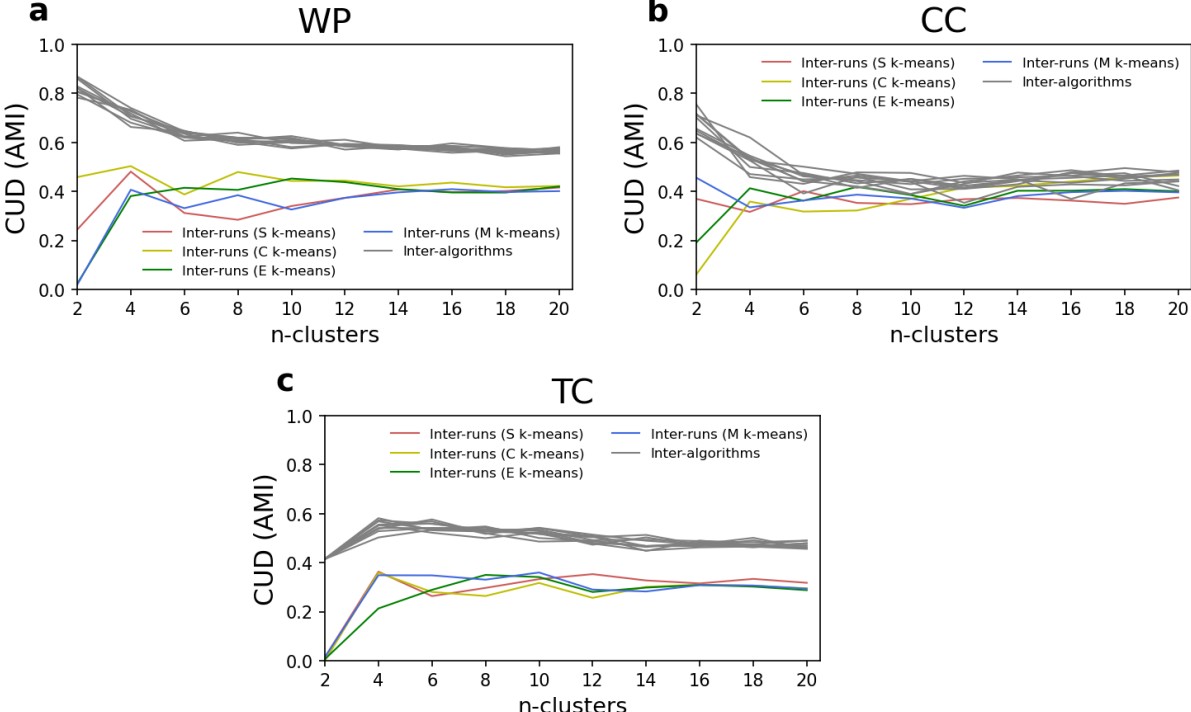

**Fig. 10 Clustering uncertainty degree (CUD) based on adjusted mutual information (AMI) between the clustering results from different runs (10 runs indicated by R0, R1, …, R9) of different *k*-means algorithms, i.e., S, C, E, and M *k*-means (rows), for different demo experiments: WP, CC, and TC (columns). Note that the results are from the configuration $k = 2, 4, …, 20$.**
