# Peer review of "Structural *k*-means (S *k*-means) and clustering uncertainty evaluation framework (CUEF) for mining climate data"

_Geoscientific Model Development, 2022_

## Author Comment (AC1)

*Note that this is the responses to the reviewer 1's comment. The revised manuscript is not included here, because it is not required in this stage.*

**Response to Reviewer's comment**

Manuscript by Doan et. al. presents a S k-means clustering framework, improving on standard k-means clustering, and demonstrate their application to several climate datasets.

Manuscript presents a methods focused study, which however lacks sufficient discussion to demonstrate the benefits of the proposed algorithmic improvements to standard k-means algorithm. Section "Results and Discussions" focus more on Results and less on Discussion, which is the critical weakness of the manuscript in its current form.

We appreciate the reviewer for his/her critical, and insightful comments, which are very helpful in improving this manuscript. We have addressed all the comments point-by-point adding appropriate discussions, some of which are based on current results and some on additional tests and analyses. In summarization, additional discussions are to address:

a)  how can S k-means capture the "structuredness" of input data,
b)  uniqueness, and new insight that S k-means enables, quantified by the Shannon entropy,
c)  the novelty of the clustering uncertainty evaluation in a broader context,
d)  additional explanations regarding methods.

We hope the reviewer satisfy with the responses.

1. Manuscript is missing several key references from the reference list.  Wang et. al. 2004, Wang and Bovik, 2009 Mo et al., 2014; Han and Szunyogh, 2018; Doan et al., 2021

We have added these references to the revised manuscript.

2. One of the motivation for the proposed work, as discussed in introduction, is to mine the unique "structuredness" of temporal and spatial climate data (Line 67-81). However, rest of the manuscript focused on comparison of various clustering methods based on Silhouette scores, uncertainty degree etc. Proposed S k-means consistently shows better scores than the other methods, but if and how it better captures the "structuredness" of the data need to be discussed, since that's the key contribution of the study.

We agree. We have added discussion on how S k-means captures the "structuredness" of the data into the manuscript. We focus on the ability of the algorithm to distinguish the difference between "imagination" data, which are generated for intuitive comprehension. An additional figure (Figure R1, which corresponds to Figure 8 in the revised manuscript) has also been added to the manuscript.

"To understand how S k-means clusters data, it is essential to see how the algorithm recognizes the similarity between objects. For intuitive comprehension, we generated "imagination" two-dimensional air pressure patterns and showed them in Figure 8. In the figure, the reference pattern (a) illustrates two air pressure extrema (Low and High) located symmetrically on the left and right sides. Other patterns for comparison are Gaussian noise contamination (b), blurring (identically distributed pattern) (c), luminance shift (d), contrast stretch (e), and spatial shift (f). The patterns other than the reference are intentionally generated so that those ED (Euclidean distance) to the reference are identical (=2.9). With S-SIM, the similarities are ranked in order: S-SIM(d, a) = .99 > S-SIM(e, a) = .8 > S-SIM(b, a) = .67 > S-SIM(f, a) = .5 >> S-SIM(b, a) = 0. This example demonstrates well the ability of S-SIM in recognizing the difference between two-dimensional patterns, which ED cannot do. More interestingly, the similarities ranked by S-SIM fits well with human perception. For example, the similarity between c and a is 0 (no similarity). With using S-SIM, S k-means can avoid by-chance centroid assignments, which ED-based k-means might not. Though this example shows the two-dimensional data, it could be the same for the time series, where a temporal instead of a spatial relationship characterizes the structuredness of data."

[Figure]

*Figure R1. Imagination air pressure patterns. Subpanels are the reference (a), Gaussian noise contamination (b), blurring (to mean value) (c), luminance shift (d), contrast stretch (e), and spatial shift (f). The ED (Euclidean distance) and S-SIM (structural similarity) values shown above each panel are those calculated to reference one (a). The rightmost subpanel shows the cross-session (between two points P1 and P2 in a)) with L, H indicating the location of imagination Low and High air pressure extrema.*

3. Structural similarity metric (Section 2.2) is the most important part of the study. However, several symbols/terms in equations 2, 3 and on lines 142-145 are not defined or explained. In particular the equations for luminance, contrast and structure. And the cited articles (Wang et. al. 2004, Wang and Bovik, 2009) that developed the similarity metrics are missing from the reference list. That makes it difficult to understand the similarity metric. Aside from describing

equations for S-SIM, there are disussions, in methods section or later, as to how these structural metrics capture the spatial and temporal structuredness of climate data.

We have added more the explanation for equations 2, 3 clarifying the concept of luminance, contrast and structure similarities. All symbols/terms and notations have been checked to assure that they are all appropriated defined. Regarding the second part of the question, we believe that it is appropriately addressed in the above response to the previous comment. We kindly ask the reviewer to go back to check it.

4. Discussion of clustering results in Section 5.2 is very high level.  Question remains, aside from slightly higher scores what unique and new insights does the S k-means clustering enabled?

We have added new insight into S $k$-means clustering results, focusing on its unique characteristics. To support the arguments, we use the Shannon entropy concept and calculate clustering entropy which is shown in Figure R2 (corresponding to Figure 7 in the manuscript). Following are the details of the discussion, which have been also added in the revised manuscript.

"Insight into the uniqueness of S k-means has great practical implications/instructions for ending users. As explained above, clustering patterns alone are significant only after the physical meaning is assigned or used for a practical purpose like a prediction. Doing either does not fall into the scope of this study (it is a huge work and must be addressed in an independent study). Here we adopt another approach to discuss the S $k$-means performance. Looking carefully at Figure 3, one might realize an anomaly of S k-means compared with the others in Silhouette plots. S k-means is likely to generate what we call "high-ordered" clustering, i.e., one dominant weather pattern (the larger group size) besides several non-dominant ones (the smaller group size). The same trend is consistent with different k settings. This finding agrees well with prior knowledge of Japanese winter weather patterns. Recall that winter in Japan, due to its specific location, is characterized by dominated winter-type pattern (Low in the east and High in the west), which is well recognized by the meteorological research community and local people. The finding leads to hypotheses: (i) Does S k-means perform clustering closer to human perception than other algorithms? (ii) Is it an intrinsic property of S k-means that tends to generate "highly-ordered" clustering?

To examine the hypotheses mentioned above, we adopt Shannon entropy to quantify the "orderliness" of clustering (see Method part for more details). The results, illustrated in Figure 7, show a good agreement with calculated clustering entropy values with human intuition. That is, lower entropy (highly ordered clustering) is consistently seen in S k-means, rather than other algorithms, for the WP experiment (Fig. 7a). But this trend is not confirmed for other experiments like CC, and TC (Fig. 7b, c). Because we did not ensure it for CC and TC, we can eliminate the second hypothesis, i.e., "highly-ordered clustering" is not an intrinsic property of S k-means. Now we have the first hypothesis remaining, i.e., S k-means is likely to generate clustering closer to human perception. Note it is still too early to conclude about the superiority of S k-means based on only

what is shown here. More investigation with a wide range of data types (with well-defined prior knowledge) is needed to gain conclusive insight into the algorithm."

[Figure]

*Figure 2. Shannon entropy of clustering results. Comparison of the average silhouette score (S-score) of S, C, E, and M k-means for k = 2, 4, ..., 20 for three demonstration experiments: WP (a), CC (b), and TC (c). The uncertainty range in each line indicates the standard deviations of the scores among ten runs with randomized initializations. (**Figure 7 in the manuscript**)*

5. I am glad to see S k-means being compared with three other k-means variants. They were all run for a 11 different 'k' and with 10 random ensembles each, resulting in a toal of 1320 clustering runs. BUT were all four k-means variants run with exactly the same random starting centroids for the purpose of comparison? It's important to do that for a fair comparison. Also, was a consistent convergence criteria used for all four methods? Converge criteria was mentioned on Lines 128-129, but what criteria was used in the study never discussed.

The four k-means variants have been conducted with the randomized centroids each time. It is because we aimed to compare four algorithms as they are as integrated systems. However, we also understand well the concern of the reviewer. To address the reviewer's concern and assure that our conclusion in the original manuscript is robust, we have run additional experiments assuming the same random starting centroids. In detail, the extra 132 run (3 experiments x 11 $k$ settings x 4 $k$-means variants) has been conducted based on 33 pre-defined starting centroid sets (3 experiments x 11 $k$ settings).

The quick conclusion is the uncertainty (related to clustering algorithm selection) remains even though the same starting centroids are used. Compare Figure 3 (additional runs) and Figure 4 (original runs) for $k$ = 4; the CUD in clustering results from the four $k$-means variants is confirmed at the same level regardless the fully randomized initialization, or identical predefined centroid assumptions. It highlights that the similarity recognition scheme dominantly causes the uncertainty associated with selecting the k-means variants.

The results demonstrated the validity of the original comparison. For this reason, we do not change the structure of the original paper. However, we added a few discussions about this.

"Note that the four k-means variants run with the randomized centroids each time. It is because we aimed to compare four algorithms as they are as integrated systems. Additional runs using the same starting centroids for k-means variants show that the uncertainty related to clustering algorithm selection remains regardless of using the same starting centroids or randomized initialization."

[Figure]

Figure 3. Clustering uncertainty degree (CUD) based on adjusted mutual information (AMI) between clustering results from different k-means algorithms, i.e., S, C, E, and M k-means, for different demo experiments: WP, CC, and TC. (a, b, c) CUD in heatmaps, and (d, e, f) visualization of the interconnection using the chord diagrams. Note that the results are from the configuration with k=4 and **the four k-means variants use the same starting centroids.**

[Figure]

*Figure 4. (**Figure 10 in the manuscript**) Clustering uncertainty degree (CUD) based on adjusted mutual information (AMI) between clustering results from different k-means algorithms, i.e., S, C, E, and M k-means, for different demo experiments: WP, CC, and TC. (a, b, c) CUD in heatmaps, and (d, e, f) visualization of the interconnection using the chord diagrams. Note that the results are from the configuration with k=4 and **the four k-means variants use the randomized starting centroids**.*

Regarding the second part of the reviewer's comment, the consistent convergence criterion has been used for all four methods. We have added this information to the revised manuscript.

> "Technically, the algorithm converges if the sum of the mean square errors of centroids versus those in the previous step becomes zero. The convergence criterion is the same for all k-means variants used in this study. An iteration limitation is set up to 100 to avoid the infinite loop of iterations."

6. Lines 364-365 "As the first study to address this issue, we believe that CUEF can constitute a new standard for addressing uncertainty issues when performing data clustering in (but not limited to) climate science." -- This is an overstatement. It's well know that custering algorithms are local search methods that are sensitive to random start, however, there are number of approaches in published literature to identify good seeds and ensure that algorithms can converge to a consistent cluster set.

We partially agree with this comment. A reason is that the clustering uncertainty evaluation framework (CUEF) must be understood in **a broader context**. The clustering uncertainty is not only caused by how the algorithm is initialized. It is also caused by selecting different *k*-means algorithms, or different clustering algorithms other than *k*-means such as affinity propagation, DBSCAN, self-organizing map, etc. It is also caused by input data (for example, we confirmed that the uncertainty in clustering Japanese summer weather patterns is much higher than that we cluster winter weather patterns though not shown here).

Nevertheless, we agree that there are number of approaches to identify good seeds to improve the convergence of $k$-means. According to knowledge of the authors, the most well-known approach is $k$-means ++ (Arthur and Vassilvitskii, 2007). The intuition behind this method is that spreading out the $k$ initial cluster centroids is preferable: the first cluster centroid is chosen randomly from the input data points. Each subsequent cluster centroid is determined from the remaining input data points with probability proportional to its distance from the point's closest existing centroid. One might note that $k$-means ++ still, literarily, relies on random selection of the first "seed". For that the final "seed" set will be different from each other resulting the uncertainty in the clustering results. Also, note that random choice of seed is not bad assumption, rather it is necessary to avoid the bias of specified seeds.

To demonstrate it in more clear and evident way, we have run additional simulations, in which the **$k$-means++** scheme of initialization is used instead of fully randomized method in the original $k$-means algorithm. The results are shown in figures following demonstrate two things.

(1) Obviously, clustering uncertainty in using different $k$-means ++, i.e., S, C, E, M $k$-means++ still exists (Fig. R3). **The degree of uncertainty is the same** with that among S, C, E, M $k$-means (as shown in Fig. R4 which corresponds to Fig 9 in the original manuscript).

(2) Clustering uncertainty exists among different runs (Fig. R5). It is because $k$-means ++ is not free from random choice of seed. However, interestingly and also somehow expectedly, the clustering uncertainty caused by $k$-means++ is smaller than original $k$-means (Fig. R7 – Figure 10 in the manuscript). In TC experiment, E $k$-means ++ can provide a zero uncertainty. In WP experiment, the uncertainty is higher, implies that it depends much on data used.

We have added additional discussion about CUEF into the revised manuscript:

"Our proposed clustering uncertainty evaluation framework (CUEF) must be understood in a broader context. The clustering uncertainty can be caused by selection of clustering algorithms (other than $k$-means such as affinity propagation, DBSCAN, self-organizing map, etc.), by initialization scheme, and by input data itself. Focusing on initialization scheme for k-means, one might note that that there are number of approaches to identify good seeds to improve the convergence of final outcomes. The most well-known approach is k-means ++ (Arthur and Vassilvitskii, 2007). The intuition behind this method is that spreading out the k initial cluster centroids is preferable: the first cluster centroid is chosen randomly from the input data points. Each subsequent cluster centroid is determined from the remaining input data points with probability proportional to its distance from the point's closest existing centroid. One might note that k-means ++ still, literarily, relies on random selection of the first "seed". For that the final "seed" set will be different from each other resulting the uncertainty in the clustering results. Also, note that random choice of seed is not bad assumption, rather it is necessary to avoid the bias of specified seeds.

To see how uncertainty problem is solved with $k$-means ++, we have run additional simulations, in which the $k$-means++ scheme of initialization is used instead of fully

randomized method in the original *k*-means algorihm. The results (shown in the appendices) demonstrate that inter-algorithm clustering uncertainty with *k*-means ++ still exists. The degree of uncertainty is the same with that among original *k*-means variants (Fig. 9). Interestingly and somehow expectedly, the inter-run uncertainty with *k*-means++ is smaller than original *k*-means, implying that improving initialization could reduce the uncertainty of clustering results, though this depends much on the type of input data. "

[Figure]

*Figure 5. Clustering uncertainty degree (CUD) based on adjusted mutual information (AMI) between clustering results from different k-means algorithms, i.e., **S, C, E, and M k-means++**, for different demo experiments: WP, CC, and TC. (a, b, c) CUD in heatmaps, and (d, e, f) visualization of the interconnection using the chord diagrams. Note that the results are from the configuration with k=4 and the first initialization run.*

[Figure]

*Figure 6 (**Figure 10 in the manuscript**) Clustering uncertainty degree (CUD) based on adjusted mutual information (AMI) between clustering results from different k-means algorithms, i.e., **S, C, E, and M k-means**, for different demo experiments: WP, CC, and TC. (a, b, c) CUD in heatmaps, and (d, e, f) visualization of the interconnection using the chord diagrams. Note that the results are from the configuration with k=4 and the first initialization run.*

[Figure]

*Figure 7 Clustering uncertainty degree (CUD) based on adjusted mutual information (AMI) between the clustering results from different runs (10 runs indicated by R0, R1, ..., R9) of different **k-means++ algorithms, i.e., S, C, E, and M k-means++** (rows), for different demo experiments: WP, CC, and TC (columns). Note that the results are from the configuration with k=4 and the first initialization run.*

[Figure]

*Figure 8 (**Figure 11 in the manuscript**) Clustering uncertainty degree (CUD) based on adjusted mutual information (AMI) between the clustering results from different runs (10 runs indicated by R0, R1, ..., R9) of different **k-means algorithms, i.e**., **S, C, E, and M k-means** (rows), for different demo experiments: WP, CC, and TC (columns). Note that the results are from the configuration with k=4 and the first initialization run.*

**Reference:**

- Arthur, D.; Vassilvitskii, S. (2007). "k-means++: the advantages of careful seeding" (PDF). Proceedings of the eighteenth annual ACM-SIAM symposium on Discrete algorithms. Society for Industrial and Applied Mathematics Philadelphia, PA, USA. pp. 1027–1035.

7. Lines 370-374: "This makes sense because different data have different topologies, which can make them unsuitable or even invalid for a clustering solution. The question of whether it is valid or meaningful to apply a clustering solution to a dataset is more important than how to find the best method of clustering. Although this issue is fundamentally important, to the authors' best knowledge, no studies have addressed this question or proposed a solution, at least among the climate sciences." -- this again is broad and biased inference based on the demonstrated applications and results.

We agree that the statement could sound overestimation, though we have a reason to say that. In this study we are trying to raise the awareness of clustering application research community (at least limited to climate science, where we are accustomed) that "**do right things**" is better than "**do things right**". Before solving a clustering problem (whose purposes could be to gain knowledge or to do prediction), a researcher needs to ask the first question whether it is meaningful to apply clustering approach, i.e., is this right thing to do? If the intrinsic uncertainty in the problem is too large one have to give this method up because the results even obtained are not robust enough, and discussion based on these will be misleading.

In the revised manuscript, we have revised the text in Lines 370 – 374 to, we reduce the "Although this issue is fundamentally important, to the authors' best knowledge, no studies have addressed this question or proposed a solution, at least among the climate sciences."

To

> "In the other words, this study attempts to raise awareness of clustering-application community that "before trying to do thing right, one must know it is right thing to do"."

8. Authors have termed their clustering framework to be novel, including in the title of the manuscript, which in my opinion is overstated and not justified. There are three key methodology elements in the paper + application to three select climate datasets.

Application component of study is weak and limited in scope. But author's acknowledge that application/interpretation was not the focus of their study, Lines 277-278 "We do not intend to physically interpret the specific clustering outcomes, although some phenomenal explanations are provided in the manuscript." So novelty is not in the three applications.

Three elements of methodology are adopted from published literature:

1. Structural similarity based k-means -- adopted from Wang et. al. 2004, Wang and Bovik, 2009

2. Evaluation of clustering algorithms using Similarity distributions (adopted from Doan et. al. 2021), Silhouette scores (adopted from Hassani and Seidl, 2017).

3. Clustering uncertainty degree and information theory (Vinh et al. (2009))

Building upon published literature is normal discourse of scientific research. But I suggest reconsidering the use of term "novel".

The reviewer is correct about basic structure of this study. We sincerely accept the request of the reviewer to reconsider the use of term "novel" in the title. We have revised the title to

> "Structural k-means (S k-means) and clustering uncertainty evaluation framework (CUEF) for mining climate data".

We also understand that the concept of "novelty" is usually subjective depending on standing point of viewer. Apart from discussion whether it is proper to have "novelty" in the title, let us remind the reviewer about some "new values" that we added to current literature. First, S k-means algorithm is the first variant, according to knowledge of the authors, adopts **structural similarity paradigm** to cluster things. There have been a lot of variants of k-means regarding how to determine similarity/distance between objects, but most are based on distance paradigm.

The second new value is the clustering uncertainty evaluation framework. The reason we call it a **framework** because it is more than an application of a technique like mutual information. We propose the way to evaluate the meaningfulness of application of clustering solution for a given problem. We use mutual information as a showcase, though we can use different criteria such as "rand index", which has been developed for the same purpose as mutual information. Also, remember that the adjusted mutual information (Vinh et al., 2009) is primarily developed to measure the "goodness" of clustering algorithm based on assumption of existing "ground truth". Here we **diversify** the primary purpose by using it to evaluate the uncertainty/consistent/convergence of a clustering solution. So, using mutual information have to be understood as showcase of CUED, but not CUED itself.

We have added to revised manuscript.

> "Note that the CUEF proposed in this study it is more than an application of a technique like mutual information. The idea is to propose the way to evaluate the meaningfulness of applying a clustering solution for a given problem. Here we used mutual information as a showcase. Recall that the adjusted mutual information (Vinh et al., 2009) is primarily developed to measure the "goodness" of clustering algorithm versus prior known "ground truth". Here we diversify the primary purpose by using it to evaluate the uncertainty/consistent/convergence of a clustering solution. Exactly saying, here we can use different techniques to do so, for rand index. So, using mutual information have to be understood as showcase of CUED, but not CUED itself."

---

## Author Comment (AC2)

*Note that this is the responses to the reviewer 2's comment. The revised manuscript is not included here, because it is not able to be uploaded in this stage.*

**Response to Reviewer 2's comments**

This study by Doan et al. presents the use of a S k-means clustering as a better alternative for climate and atmospheric science to clustering data than traditional k-means methods. This study introduces a novel framework to identify uncertainty within clustering methodologies and said framework introduces a methodology by which researchers can compare different clustering techniques with each other in a way that doesn't require a ground truth dataset to exist by which to compare results to. The study presents the methdology in an excellent manner that seems like it would be easy to replicate/apply to future studies.

S k-means is a useful technique that adapats SSIM techniques, traditionally used in image comparison analysis, to be applied to climate data. It is an improved technique, compared to the traditional distance metric comparisons, as this takes into account both spatial and temporal differences in datasets. This manuscript does a good job at summarizing the use of the aforementioned techniques with respect to three example tests for typical climate situations in which clustering is used. **However, this manuscript lacks in the discussion and summary sections. The manuscript needs to emphasize more as to the usefulness of this new uncertainty framework compared to current available methodology. The results are well explained, but there is a lack of discussion about how this brings a significant change to current techniques/how this improves current understanding and techniques.**

Thanks, the reviewer for the positive feedbacks. We agree that the discussion about the significant contribution of the proposed methodologies versus current techniques is needed to improve the quality of the manuscript. Following the advice of the reviewer, we add following discussion into the revised manuscript. The summary section is revised accordingly. We hope the reviewer satisfy with this revision.

> "Another benefit of CUEF is that it can measure the meaningfulness of clustering given data. To date, clustering algorithms including *k*-means have been used primarily to either explore unknown atmospheric patterns or support predictions. The most common approach is using clustering techniques within the framework of "detection-and-attribution", i.e., detect specific atmospheric events, e.g., abnormally hot weather or heavy precipitation, then attribute the causes to atmospheric regimes/patterns revealed by clustering analysis (Esteban et al., 2005; Houssos et al., 2008; Spekat et al., 2010; Zeng et al., 2019; Smith et al., 2020). Clustering techniques are also used for weather forecasts or climate predictions (Kannan and Ghosh, 2011; Gutiérrez et al., 2013; Le Roux et al., 2018; Pomee and Hertig, 2022) or for reconstructing historical data (Camus et al., 2014).
>
> No doubt, clustering analysis largely contributes to advancing climate sciences alongside other data analysis and numerical modeling techniques. The essence of the technique lies in its ability to extract knowledge (patterns) from data. It allows researchers to

discover unseen structures hidden in data which is massive and inaccessible to human perception. So far, tremendous efforts have been invested in either proposing/improving clustering algorithms or inventing criteria for evaluating the goodness of the results. Such efforts could be classified as "attempts to do things right." A question posed here is more fundamental in the sense of how to justify the selection, i.e., "whether it is the right thing to do (the right method to select)?". This study proposes a quantitative framework in which the users could justify the selection directly based on the data rather than relying on the literature review (select it because other researchers use it). Such kind of justification is more or less a fallacy due to the diversifying clustering problems in climate science, and the variety of clustering algorithms. Also, climate data, whose types and amounts are increasing at an unprecedented pace, is adding challenges to experience-based justification. According to the authors' knowledge, the CUEF is the first attempt to address this issue. Though there is still free room for further development, CUEF is believed to constitute a new standard for climate data clustering. We recommend CUEF as a necessary procedure before applying clustering techniques. Even though the justification might depend on multiple factors other than the data-oriented uncertainty, such as how the clustering results will be used in further analysis processes, CUEF can support the explanation and discussion of the clustering results."

1. Table 1 provides a nice summary of different metrics compared between the different k-means models used in the study. In the text, the mean and standard deviation are mentioned from the table, however, the other metrics are not mentioned at all other than in passing. The Shannon metric needs to be explained more and some presentation of the data should be given in the text to give the reader some context as to its meaning and how it is used in this study.

Thank you for the noticing. In the revised manuscript, we have added the explanation related to other metrics. We copied Figure 2 and Table 1 here together with additional explanation for reference.

> "Before analyzing the $k$-means clustering results, we diagnosed the nature of the input data using S-distributions (or S-D). S-Ds provide "global" insights into how data vectors are related to each other in four S-SIM, COR, ED, and MD topological spaces. The results, which are shown in **Figure 2**, demonstrate an apparent difference in the shape of the S-Ds. Notably, the S-Ds for ED and MD appeared more symmetrical than those for S-SIM and COR across the three types of input data, that is, WP, CC, and TC. For S-SIM and COR, S-Ds tended to be more tailed (both sides), with skewness over the left tail. Quantitively, the standard deviation of S-Ds for S-SIM and COR tended to be higher (0.13 – 0.20) than those for ED and MD (approximately 0.11 – 0.13) (Table 1), despite an exception for ED in the TC simulation. The skewness that measures the symmetry of S-Ds shows negative values, meaning the left-skewed distributions. Those values in Table 1 are consistent with visualization in Figure 2. Especially, S-SIM and COR tend to be higher skewed than that of ED and MD particularly in the CC and TC experiments. The consistent skew-over-left of S-SIM and COR indicates that those tend to project

"hierarchical affinity" of input vectors, meaning that a given vector tends to be closer to a certain group of peers and relatively far from another group located at the opposite end of similarity spectrum. In this sense, these results demonstrate that the discrimination ability of S-SIM and COR is higher than that of traditional distance metrics, such as ED or MD. In addition, kurtosis and Shannon entropy measure the flatness and "information value" (or "information gain" in the case of comparison), respectively, of S-Ds. Overall, kurtosis values are consistent with visualized results in Figure 2, i.e., S-Ds of S-SIM and COR tend to spread more over two tails compared with ED and MD. Entropy, on the other hand, does not show obviously higher and lower trends of S-SIM, and COR compared with ED and MD, and it is likely more data dependent."

[Figure]

**Fig. 2 (in the manuscript) Comparison of the S-distributions of normalized pairwise similarity using the structural similarity (S-SIM), the Pearson correlation coefficient (COR) the Euclidean distance (ED) and the Manhattan distance (MD) for three demonstration experiments: WP, CC, and TC. With a population size of N, $\frac{N(N-1)}{2}$ values of pairwise similarity are observed because S-SIM, COR, ED and MD are symmetric measures and self-similarity is excluded. Values are normalized from 0 to 1. The maximum similarity is 1, which corresponds to completely similar, and the minimum similarity is 0, which corresponds to the lowest pairwise similarity.**

**Table 1 (in the manuscript). Statistical metrices of S-distributions for three demonstration input datasets, i.e., weather pattern (WP), climate change (CC), and tropical cyclone (TC). The different distance/similarity measures are structural similarity (S-SIM), the Pearson correlation coefficient (COR), Euclidean distance (ED) and Manhattan distance (MD). Statistical measures include the mean (Mean), standard deviation (STD), skewness (SKEW), kurtosis (KUR) and Shannon entropy (ENTROPY)**

|  | WP | | | | CC | | | | TC | | | |
| --- | --- | --- | --- | --- | --- | --- | --- | --- | --- | --- | --- | --- |
|  | S-SIM | COR | ED | MD | S-SIM | COR | ED | MD | S-SIM | COR | ED | MD |
| Mean | 0.68 | 0.71 | 0.67 | 0.68 | 0.71 | 0.81 | 0.66 | 0.65 | 0.81 | 0.87 | 0.65 | 0.69 |
| STD | 0.18 | 0.19 | 0.11 | 0.11 | 0.20 | 0.13 | 0.12 | 0.13 | 0.14 | 0.11 | 0.15 | 0.13 |
| SKEW | -0.66 | -0.81 | -0.73 | -0.74 | -1.08 | -1.25 | -0.65 | -0.67 | -1.10 | -1.67 | -0.46 | -0.59 |
| KUR | -0.18 | 0.00 | 0.58 | 0.64 | 0.97 | 1.79 | 0.59 | 0.58 | 1.15 | 3.31 | -0.32 | 0.03 |
| ENTROPY | 2.83 | 2.79 | 2.19 | 2.16 | 2.83 | 2.29 | 2.32 | 2.36 | 2.30 | 1.80 | 2.57 | 2.45 |

2. Many references from the text are missing citations. Please check over the references in the paper to make sure all are cited, here are a few that I found that were not cited: Jancey 1966, Lloyd 1957, Wang et al. 2004, etc.

We apology for this inconvenience caused for the reviewer. We will add these references into the revised manuscript.

3. This study intends to establish both the uncertainty framework and the s k-means methodology as a new standard for data mining in the climate sciences. While the uncertainty framework definitely provides a new standard by which to test the usefulness and effectiveness of different clustering algorithms against each other, no work has been shown as to the ability of the s k-means clustering. While comparisons are shown between the s k-means to other k-means clustering measures, **we cannot objectively say from this study that the S k-means method better captured the underlying structures** within the data compared to the other k-means models. **A more comprehensive case study would be needed, rather than the short test cases, that applies the methodologies to a known problem that has a ground truth that can be compared back to.**

The reviewer is very critical on this point. "Ground truth", if exists, is the best solution to determine the goodness of one method against another. However, there are reasons that we do not use "ground truth" in this study. First, we have no reliable "ground truth", i.e., "real" patterns of three datasets, WP, CC, and TC. The reviewer might notice that the lack of "ground truth" is common in other atmospheric data also, not only related to our experience settings. Because it is difficult to define "true" weather pattern, even though some individuals (i.e., weather forecasters) might claim that they have. In our opinion, climate data is very, that we called, "contextual" data, i.e., a claim of a weather pattern (or typhoon pathway pattern) is exclusively data dependent, it is rather associated with broader contexts of personal experiences, knowledges. It is why we try to avoid "personal-experience involvement" in the evaluation until "universal ground truth" is available.

Nevertheless, we add discussion about the ability of S $k$-means in capturing the "structuredness" of the data with additional analysis and plotting (show below) to address different aspect of the reviewer's comment. Here we focus on the ability of the algorithm to distinguish the difference between objects. With comparing imagination weather patterns (generated for intuitive comprehension) we demonstrate that using S-SIM could provide better (closer to human intuitive) similarity recognition than distance metrices. The following

discussion and the additional figure (Figure 8 in the revised manuscript) are added to the manuscript.

"To understand how S $k$-means clusters data, it is essential to see how the algorithm recognizes the similarity between objects. For intuitive comprehension, we generated "imagination" two-dimensional air pressure patterns and showed them in Figure 8. In the figure, the reference pattern (a) illustrates two air pressure extrema (Low and High) located symmetrically on the left and right sides. Other patterns for comparison are Gaussian noise contamination (b), blurring (identically distributed pattern) (c), luminance shift (d), contrast stretch (e), and spatial shift (f). The patterns other than the reference are intentionally generated so that those ED (Euclidean distance) to the reference are identical (=2.9). With S-SIM, the similarities are ranked in order: S-SIM(d, a) = .99 > S-SIM(e, a) = .8 > S-SIM(b, a) = .67 > S-SIM(f, a) = .5 >> S-SIM(b, a) = 0. This example demonstrates well the ability of S-SIM in recognizing the difference between two-dimensional patterns, which ED cannot do. More interestingly, the similarities ranked by S-SIM fits well with human perception. For example, the similarity between c and a is 0 (no similarity). With using S-SIM, S k-means can avoid by-chance centroid assignments, which ED-based k-means might not. Though this example shows the two-dimensional data, it could be the same for the time series, where a temporal instead of a spatial relationship characterizes the structuredness of data."

[Figure]

**Figure 8 (in the revised manuscript). Imagination air pressure patterns. Subpanels are the reference (a), Gaussian noise contamination (b), blurring (to mean value) (c), luminance shift (d), contrast stretch (e), and spatial shift (f). The ED (Euclidean distance) and S-SIM (structural similarity) values shown above each panel are those calculated to reference one (a). The rightmost subpanel shows the cross-session (between two points P1 and P2 in a)) with L, H indicating the location of imagination Low and High air pressure extrema.**

4. The use of 3 different test case scenarios to test the uncertainty framework was a great idea and well presented. It gives good insight into how this methodology can be used in the wide-array of applications in climate science.

Thanks the reviewer for this compliment.

5. Lines 370-374. This question of applying the framework to see whether data is suitable for clustering is a much more novel approach and useful to the science than comparing the initializations. There are many other methodologies and ways to get suitable initializations for clustering and help datasets to converge on useful clustering.

Thank you. Indeed, our study emphasize the effectiveness of the CUED for when comparing algorithms and datasets. Though initializations could cause the uncertainty but some improvement such as k-means++ could help to reduce uncertainty and preserve the consistency in clustering results. The discussion regarding this comment can be found the above answer (to the general comment).

6. Lines 370-374. It is tough to say with respect to WPs that clustering my be ineffective. WPs present a lot of uncertainty compared to other types of climate data, so without care as to what is being analyzed/searched for in the data, uncertainty analysis may present false positives for datasets that would not be suitable for clustering. This isn't a problem with the methodology, the authors do note that these are inherently a data issue, which this methodology does not take into account. The authors could do to make note of similar situations in the manuscript for those who would use this method in the future.

The reviewer is correct. We add some clarification to avoid the potential misinterpretation to the revised manuscript.

> "Note that CUEF provide a method to quantify the uncertainty/consistency of clustering solutions from the data science aspect. However, the decision whether to adopt clustering techniques could depend on another factor, such as how the results will be used and interpreted. In such a case, CUEF could be used to support explanation regarding the robustness or the clustering results."

7. Some figures need revision, specificaly figures 3, 4, and 5. In Figure 3, the silhouette score charts are very small compared to the WP plots. Make them a similar size and make the text size more legible. Figures 4 and 5 have the silhouette score charts inside of the other figures. There is far too much going on inside these figures as it is, and adding the silhouette plots inside here makes it more cluttered and confusing to understand. Move them outside the plots and enlargen them.

We have replotted the Figures 3, 4, and 5 exactly following the suggestions of the reviewer. The replotted figures are attached below for reference.

[Figure]

**Fig. 3 (in manuscript). The silhouette score charts become bigger and text size more legible.**

[Figure]

**Fig 4 (in the revised manuscript). Moved the silhouette score charts outside of maps and enlarged the charts and made the text size legible.**

[Figure]

**Fig 5 (in the revised manuscript). Moved the silhouette score charts outside of maps and enlarged the charts and made the text size legible.**

**Minor notes:**

Lines 72-74: Rephrase the wording, it is confusing in this state.

We have rephrased the sentence from

> "For these reasons, k-means under the distance paradigm treats the features of the input data equally, thus mask the similarity recognition between data, consequently deteriorating the clustering outcomes."

to

> "Thus, the distance measures, which treat the features of the input objects equally, might ignore inherent "structuredness" in the objects when recognize the similarity between them. This characteristic could deteriorate the clustering outcomes."

Line 75: Remove ".It is" and use because to join the two sentences into one for better flow.

Have removed ".It is" and joined two sentences into one.

Cited Wang et al., 2004.

We used nearest-neighbor interpolation for regridding the data. We add this information into the revised manuscript.

> "The data had a horizontal resolution of 0.75°on a regular grid but were re-gridded to an equal-area scalable earth-type grid at a spatial resolution of 200 × 200 km using nearest-neighbor interpolation method."

Line 238: Could explain cluster realization better/earlier. Explaining it in this sentence while also introducing a new concept could cause confusion to the reader.

Have rephrased the text for clarification.

> "In this study, mutual information is applied to evaluate the agreement between two clustering realizations (label assignments of $N$ objects). To do so, the mathematical formula for mutual information $I(U,V)$ between two clustering realizations $U$ and $V$ is defined as follows:"

Line 239-240: What do you mean by partition set? Is this the same thing as the cluster realization?

Yes, partition set is detailed form of cluster realization. We have revised the sentence for clarification.

> "Entropies of clustering realizations are defined as the amount of uncertainty for partition sets of each realization."

Line 246: What do you mean by weakness? Is it related to the randomess you discuss in the next few lines?

Yes, mutual information is weak with random clustering (or chance). We have revised the text for easier understanding.

> "However, mutual information is weak against chance."

Line 297: Change tense of "were" to "are".

We have revised the text accordingly:

> "These regional differences are well captured by $k$-means clustering. For example, the northern part (Hokkaido) is consistently separated from other regions in terms of temperature warming."

Line 316: What does "completed by C K-means" mean? Is it a typo?

It is "competed" not "completed". We have revised the text to:

> "The performance of S $k$-means is sometimes competed by C $k$-means."

---

## Author Response (AR1)

**Responses to Reviewer 1's comment**

Manuscript by Doan et. al. presents a S k-means clustering framework, improving on standard k-means clustering, and demonstrate their application to several climate datasets.

Manuscript presents a methods focused study, which however lacks sufficient discussion to demonstrate the benefits of the proposed algorithmic improvements to standard k-means algorithm. Section "Results and Discussions" focus more on Results and less on Discussion, which is the critical weakness of the manuscript in its current form.

We appreciate the reviewer for his/her critical, and insightful comments, which are very helpful in improving this manuscript. We have addressed all the comments point-by-point adding appropriate discussions, some of which are based on current results and some on additional tests and analyses. In summarization, additional discussions are to address:

- a) how can S k-means capture the "structuredness" of input data,
- b) uniqueness, and new insight that S k-means enables, quantified by the Shannon entropy,
- c) the novelty of the clustering uncertainty evaluation in a broader context,
- d) additional explanations regarding methods.

We hope the reviewer satisfy with the responses.

1. Manuscript is missing several key references from the reference list. Wang et. al. 2004, Wang and Bovik, 2009 Mo et al., 2014; Han and Szunyogh, 2018; Doan et al., 2021

We have added these references to the revised manuscript.

2. One of the motivation for the proposed work, as discussed in introduction, is to mine the unique "structuredness" of temporal and spatial climate data (Line 67-81). However, rest of the manuscript focused on comparison of various clustering methods based on Silhouette scores, uncertainty degree etc. Proposed S k-means consistently shows better scores than the other methods, but if and how it better captures the "structuredness" of the data need to be discussed, since that's the key contribution of the study.

We agree. We have added discussion on how S k-means captures the "structuredness" of the data into the manuscript. We focus on the ability of the algorithm to distinguish the difference between "imagination" data, which are generated for intuitive comprehension. An additional figure (Figure R1, which corresponds to Figure 8 in the revised manuscript) has also been added to the manuscript (Line 350 - 360).

"To further the discussion from a different aspect, we examine how the similarity between objects is recognized in *k*-means variants. For intuitive comprehension, we generate "imagination" weather patterns and assess the discrimination ability of similarity/distance metrices. **Figure 8** illustrates the weather patterns including the reference (a), characterized by two extrema (Low and High) symmetrically distributed over both sides, the Gaussian noise contamination (b), the blurring (to the mean value) (c), luminance shift (d), contrast stretch (e) and the spatial shift (f). Though the Euclidean distance from these patterns (b-e) to the reference are intentionally set to be identical (=2.9), by using S-SIM, one can rank the similarities with descending order: S-SIM(d-a) = .99 > S-SIM(e-a) = .8 > S-SIM(b-a) = .67 > S-SIM(f-a) = .5 >> S-SIM(b-a) = 0. This

simple demonstration confirms the superiority of S-SIM in recognizing the difference between two-dimensional patterns, agreeing well with human intuition compared to ED. This implies that S-SIM could reduce the situation of random classification (i.e., an object is assigned to a centroid by chance) adding confidence to S k-means derived results. Though this result is shown for the two-dimensional data, it is believed to be true for one-dimensional structured data like time series."

Figure R1. Imagination air pressure patterns. Subpanels are the reference (a), Gaussian noise contamination (b), blurring (to mean value) (c), luminance shift (d), contrast stretch (e), and spatial shift (f). The ED (Euclidean distance) and S-SIM (structural similarity) values shown above each panel are those calculated to reference one (a). The rightmost subpanel shows the cross-session (between two points P1 and P2 in a)) with L, H indicating the location of imagination Low and High air pressure extrema.

3. Structural similarity metric (Section 2.2) is the most important part of the study. However, several symbols/terms in equations 2, 3 and on lines 142-145 are not defined or explained. In particular the equations for luminance, contrast and structure. And the cited articles (Wang et. al. 2004, Wang and Bovik, 2009) that developed the similarity metrics are missing from the reference list. That makes it difficult to understand the similarity metric. Aside from describing equations for S-SIM, there are disussions, in methods section or later, as to how these structural metrics capture the spatial and temporal structuredness of climate data.

We have added more the explanation for equations 2, 3 clarifying the concept of luminance, contrast and structure similarities. All symbols/terms and notations have been checked to assure that they are all appropriated defined. We copied here the revised text (red color) within the relevant context for the reviewer's reference (Line 136 - 160):

"The metrics for the structural similarity (S-SIM) recognition process were first introduced by Wang et al. (2004). It was developed to better predict the perceived quality of digital television and cinematic pictures. S-SIM is intended to improve the traditional peak signal-to-noise ratio or mean squared error in detecting similarities between "structural" signals, such as images. Intuitively, S-SIM is determined by considering the differences between two input signals (vectors x, y) across multiple aspects including "luminance", "contrast", "and structure" which

represent the characteristics of human visual perception. Luminance masking is a phenomenon whereby image distortions tend to be less visible in bright regions, while contrast masking is a phenomenon whereby distortions become less visible where there is significant activity or "texture" in the image. Mathematically, S-SIM is determined as follows:

$$SSIM(x, y) = l(x, y)^{\alpha} \times c(x, y)^{\beta} \times s(x, y)^{\gamma}$$

where l(x, y), c(x, y), and s(x, y) measure similarities in luminance (brightness values), contrast and structure between sample vectors x, y with weight values  $\alpha, \beta$  and  $\gamma$ . Let  $\mu_x$  and  $\mu_y$  be the mean values;  $\sigma_x$  and  $\sigma_y$  the standard deviations;  $\sigma_{xy}$  the covariance of the two sample vectors x, y, then luminance, contrast, and structure similarities are defined as  $l(x, y) = \frac{2\mu_x\mu_y+c_1}{\mu_x^2+\mu_y^2+c_1}$ ,  $c(x, y) = \frac{2\sigma_x\sigma_y+c_2}{\sigma_x^2+\sigma_y^2+c_2}$ , and  $s(x, y) = \frac{\sigma_{xy}+c_3}{\sigma_x\sigma_y+c_3}$ . Note that  $c_1, c_2$  and  $c_3$  are parameters to stabilize the division with a weak denominator. Even if  $c_1 = c_2 = c_3 = 0$ , S-SIM still work quite well (Wang and Bovik, 2009). l(x, y) measures the similarity in brightness, i.e., the difference regarding mean values; c(x, y) quantifies the similarity in illumination variability, which is regarded to standard deviations; and s(x, y) measures the correlation in spatial inter-dependencies between images that is close to the Pearson correlation coefficient. For simplification, here we set  $c_1 = c_2 = c_3 = 0$  and weights  $\alpha = \beta = \gamma = 1$  and reduce the original formula to the following:

S-SIM
$$(x, y) = \frac{2\mu_x \mu_y \sigma_{xy}}{(\mu_x^2 + \mu_y^2)(\sigma_x^2 + \sigma_y^2)}$$

S-SIM is symmetric index, i.e., S-SIM(x, y) = S-SIM(y, x). It does not satisfy the triangle inequality or non-negativity, and thus is not a distance function. S -SIM ranges from -1 to 1, where -1 indicates totally dissimilar and 1 indicates totally similar. Wang and Bovik (2009) showed that S-SIM represents a powerful, easy-to-use, and easy-to-understand alternative to traditional distance metrics, such as Euclidean distance, for dealing with spatially and temporally structured data, i.e., data having strong spatial and temporal inter-dependencies. These inter-dependencies carry important information about the objects in the visual scene. S-SIM emerged as a "new-generation" similarity metric with an increasing number of applications outside the signal processing field, including hydrology and meteorology (e.g., Mo et al., 2014; Han and Szunyogh, 2018; Doan et al., 2021). "

Regarding the second part of the question, please see the previous response to comment 2.

**4. Discussion of clustering results in Section 5.2 is very high level. Question remains, aside from slightly higher scores what unique and new insights does the S k-means clustering enabled?**

We have added new insight into S *k*-means clustering results, focusing on its unique characteristics. To support the arguments, we use the Shannon entropy concept and calculate clustering entropy which is shown in Figure R2 (corresponding to Figure 7 in the manuscript). Following are the details of the discussion, which have been also added in the revised manuscript (Line 332 - 349).

"In pragmatic view, clustering outcomes become meaningful if they are assigned with physical meaning or successfully used for practical purposes like a prediction. Doing so does not fall into the scope of this study (it is a huge work and must be addressed in an independent study), here we adopt another approach to gain insight into the behavior of the *k*-means variants.

(1)

Give a cautious look on the silhouette plots in Figure 3, one might realize an anomaly of S *k*-means compared with the others. S *k*-means is likely to generate, say, "high-ordered" clustering, *i.e.*, one dominant weather pattern (larger group size) besides several non-dominant (smaller group size). The same trend is seen with different *k* settings (not shown). This agrees well with the prior knowledge recognized by meteorological research community and local people about the winter weather patterns in Japan (explained above. The insight leads to some possible hypotheses: (i) Does S *k*-means perform better, *i.e.*, closer to the human perception, than other variants? (ii) Is achieving "highly-ordered" clustering the intrinsic property of S *k*-means?

To examine the hypotheses, we attempt to quantify the "orderliness" of clustering outcomes using the Shannon entropy. The results, illustrated in **Figure 7**, show a good agreement between the calculated entropy values versus the intuition. S *k*-means appears to have consistently lower entropy (highly ordered clustering) than the other algorithms for the WP experiment (Fig. 7a), but not for the CC, and TC experiments (Fig. 7b, c). Because, it is not true in all the experiences, we could exclude the second hypothesis (ii), *i.e.*, achieving "highly-ordered clustering" is the intrinsic property of S *k*-means. Now hypothesis (i) remains. It is possible that S *k*-means can achieve clustering which fits closer to human perception. However, because we don't have prior knowledge regarding the CC and TC experiments, it is early to conclude that with the complete certainty. To diversify the clustering problems with different types of input data, or for different geographical areas, is necessary to obtain the comprehensive insight into S *k*-means."

---

## Author Response (AR2)

**Minor comments:**

Line 335: This sentence is worded a bit awkwardly.

We have revised the original sentence: "Give a cautious look on the silhouette plots in Figure 3, one might realize an anomaly of S k-means compared with the others." to

"By taking a careful glance at the silhouette plots shown in Figure 3, it's possible to notice a discrepancy in S k-means compared to the rest." (Line 335 – 336)

Line 336: Change besides to beside

Changed accordingly.

"S k-means is likely to generate, say, "high-ordered" clustering, i.e., one dominant weather pattern (larger group size) beside several non-dominant (smaller group size)." (Line 337)

Lines 344-345: Sentence starting with because is a bit confusing to read, had to read multiple times to understand. Could be reworded.

We have revised the original sentence: "Because, it is not true in all the experiences, we could exclude the second hypothesis (ii), i.e., achieving "highly-ordered clustering" is the intrinsic property of S k-means." to

"We can dismiss the second hypothesis (ii), which posits that achieving "highly-ordered clustering" is an intrinsic property of S k-means, because it is not universally true across all experiences." (Line 345 – 346)